# Machine learning for estimating phytoplankton size structure from satellite ocean color imagery in optically complex Pacific Arctic waters

Hisatomo Waga[1,2*], Amane Fujiwara[3], Wesley J. Moses[4], Steven G. Ackleson[4], Daniel Koestner[5], Maria Tzortziou[6], Kyle Turner[6], Alana Menendez[6], Toru Hirawake[2], Koji Suzuki[7], and Sei-Ichi Saitoh[8]

[1]International Arctic Research Center, University of Alaska Fairbanks, Alaska, USA
[2]International Polar and Earth Environmental Research Center, National Institute of Polar Research, Tokyo, Japan
[3]Institute of Arctic Climate and Environment Research, Japan Agency for Marine-Earth Science and Technology, Kanagawa, Japan
[4]Remote Sensing Division, Naval Research Laboratory, Washington, D.C., USA
[5]Department of Physics and Technology, University of Bergen, Bergen, Norway
[6]Earth & Atmospheric Sciences, City College of New York, New York, USA
[7]Faculty of Environmental Earth Science, Hokkaido University, Hokkaido, Japan
[8]Arctic Research Center, Hokkaido University, Hokkaido, Japan

*Correspondence to*: Hisatomo Waga (hwaga@alaska.edu)

**Abstract.** In response to recent advances in satellite ocean color remote sensing, we have developed a chlorophyll-a size distribution (CSD) model using machine learning (ML) approaches for optically complex Pacific Arctic waters. Previous CSD models have used principal component analysis (PCA) to retrieve spectral features from satellite-estimated phytoplankton absorption coefficient ($a_{ph}(\lambda)$) by assuming a strong correlation between the spectral features and phytoplankton size structure determined from the exponent of the CSD ($\eta$). A weakness of such approach is that it relies on satellite retrievals of $a_{ph}(\lambda)$, which can be highly uncertain due to the optical effects of water constituents other than phytoplankton. In this study, we have developed a method based on ML to use remote sensing reflectance ($R_{rs}(\lambda)$) for directly retrieving $\eta$, thus avoiding uncertainties due to the inversion of $a_{ph}(\lambda)$ from $R_{rs}(\lambda)$. Results show superior performance of the ML-based CSD models compared to the PCA-based model utilizing both $R_{rs}(\lambda)$ and $a_{ph}(\lambda)$ as predictors of $\eta$. For direct $R_{rs}(\lambda)$-based retrievals, a CSD model based on multivariable linear regression produced the best performance among all models considered. Nevertheless, models using in-situ $a_{ph}(\lambda)$ yielded better accuracy, reflecting a closer optical linkage between $\eta$ and $a_{ph}(\lambda)$ than between $\eta$ and $R_{rs}(\lambda)$. Our choice of an $R_{rs}(\lambda)$-based model for satellite application is therefore practical, motivated by the limitations and uncertainty of $a_{ph}(\lambda)$ inversions in optically complex waters. Another key finding is that more complex ML approaches do not always produce more effective models than standard linear regression. Indeed, multivariable linear regression outperformed other ML approaches for retrieving $\eta$ directly from $R_{rs}(\lambda)$, whereas support vector machine performed the best among diverse ML approaches in the case of $a_{ph}(\lambda)$. Overall, this study found benefits in using $R_{rs}(\lambda)$ with ML to improve the retrieval accuracy of $\eta$ for Pacific Arctic waters.

# 1    Introduction

Satellite remote sensing is a cost-effective tool that can provide observations across a range of temporal and spatial scales. One of the primary parameters retrieved from ocean color satellite data is the mass concentration of chlorophyll-*a* (Chl*a*; see Table 1 for symbols, definitions, and units), the primary pigment associated with photosynthesis and a key indicator of phytoplankton biomass. Satellite-derived Chl*a* observations have revolutionized our understanding of climate systems, marine ecosystems, and biogeochemical processes (McClain, 2009). However, Chl*a* alone does not provide a full description of the fundamental ecosystem functions of phytoplankton, such as nutrient uptake and cycling, energy transfer through marine food webs, deep-ocean carbon export, and gas exchange with the atmosphere (Mouw et al., 2017).

Due to the significance of phytoplankton community composition in ocean biogeochemical processes, continuous research and innovation in satellite ocean color techniques have extended our capabilities from routinely estimating Chl*a* concentration to retrieving phytoplankton functional types (PFTs) (Gordon et al., 1980; Mouw et al., 2017). PFTs are conceptual groupings of phytoplankton species that have similar biogeochemical functions (e.g., nitrogen fixers, calcifiers, dimethylsulfide producers, and silicifiers) and other characteristics such as cell size (pico-, nano-, and micro-phytoplankton). PFTs are often defined based on phytoplankton size class (PSC), phytoplankton taxonomic composition (PTC), or particle size distribution (PSD), and the choice of partitioning depends on the question at hand (Mouw et al., 2017), with no universally accepted standard (Reynolds et al., 2002). In particular, PSC serves as a useful index of the trophic state, carbon export efficiency, and productivity (Hood et al., 2006; Le Quéré et al., 2005) and, therefore, comprises the majority of PFT research.

A wide range of satellite-based methods for global estimations of PFTs have been developed to date (IOCCG, 2014). Mouw et al. (Mouw et al., 2017) provide a "user guide" for applying remote sensing techniques to monitor PFTs, explaining details of various PFT algorithms and their associated uncertainties and discussing the advantages and disadvantages of different approaches. Satellite estimation of PFTs generally exploit spectral features in remote sensing reflectance ($R_{rs}(\lambda)$), absorption coefficient of phytoplankton ($a_{ph}(\lambda)$), and/or backscattering coefficient of particles ($b_{bp}(\lambda)$) caused by variations in PFT composition (Fujiwara et al., 2011; Kostadinov et al., 2010; Li et al., 2013; Roy et al., 2017). The ocean color variables used in these spectral approaches are grouped into two categories: apparent optical properties (AOPs, e.g., $R_{rs}(\lambda)$) and inherent optical properties (IOPs, e.g., $a_{ph}(\lambda)$). Remotely sensed IOPs are derived from spectral inversion of $R_{rs}(\lambda)$ (Mobley, 1994), thereby introducing additional uncertainties for IOP-based methods compared to $R_{rs}(\lambda)$-based methods.

For global estimation of PSC, Waga et al. (Waga et al., 2017) developed a Chl*a* size distribution (CSD) model that retrieves the synoptic size structure of the phytoplankton community by determining the exponent of CSD (CSD slope; $\eta$). As opposed to other PSC approaches, $\eta$ represents the size structure of the phytoplankton community with a single value; thus, the output of the approach can be easily incorporated into ocean biogeochemical models. Akin to the PSD (Kostadinov et al., 2010; Roy et al., 2017), the arbitrariness of the arrangement of the size range is another advantage of this approach, where other methods

generally adopt a fixed target group or size class (e.g., <2 μm, 2–20 μm, and >20 μm for pico-, nano-, and micro-phytoplankton, respectively). More specifically, once $\eta$ is determined, fractional contributions of phytoplankton biomass at diverse size ranges can be estimated from $\eta$. Moreover, there is flexibility in computing $\eta$ with different combinations of size-fractionated Chl$a$, generating a comparable variable across datasets that often comprise various size ranges of size-fractionated Chl$a$ data.

The spectral features of $a_{ph}(\lambda)$ can reveal specific information regarding variations in the composition and size structure of phytoplankton assemblage (Bricaud and Morel, 1986a). For example, how pigments are distributed within a phytoplankton cell affects the magnitude of $a_{ph}(\lambda)$, while pigment composition influences the spectral shape of $a_{ph}(\lambda)$. Waga et al. (Waga et al., 2017) applied principal component analysis (PCA) to normalized $a_{ph}(\lambda)$ spectra derived from *in situ* measurements at seven wavelengths (412, 443, 469, 488, 531, 547, and 555 nm) that are consistent with spectral bands of the Moderate Resolution Imaging Spectroradiometer (MODIS). This method assumes that PCA captures spectral features of $a_{ph}(\lambda)$ as a simpler set of principal component (PC) scores while still maintaining significant patterns and trends. The relationship between $\eta$ and the resulting PC scores was then quantified by ordinary least squares regression, enabling $\eta$ to be estimated from satellite derivations of $a_{ph}(\lambda)$ (Waga et al., 2017). In order to investigate spatiotemporal variations in the size structure of phytoplankton communities and its impacts on the marine ecosystems in the Pacific Arctic, the CSD model was subsequently optimized for the Pacific Arctic based on a regional *in situ* dataset (Waga et al., 2019a). However, in Arctic coastal waters, phytoplankton absorption is typically low (only 16% of non-water absorption at 443 nm) relative to colored dissolved organic matter (CDOM) and non-algal particles (NAP) and, as a result, IOP inversion algorithms for estimating $a_{ph}(\lambda)$ are characterized by high uncertainty (Matsuoka et al., 2007). Therefore, direct approaches to estimate $\eta$ utilizing $R_{rs}(\lambda)$ may be advantageous in Arctic coastal environments, even though $R_{rs}(\lambda)$ itself is not solely influenced by phytoplankton.

The present study develops the CSD model for the Pacific Arctic utilizing diverse supervised machine learning (ML) approaches, ranging from simple linear regression to convoluted methods such as neural networks (Chen et al., 2015, 2018; Li et al., 2020, 2023; Waga et al., 2022), support vector machines (Deng et al., 2019; Selvaraju et al., 2021; Su et al., 2015), Gaussian processes (Pasolli et al., 2010), and ensemble methods (Bao et al., 2023; Qi et al., 2022; Qiao et al., 2022; Zhang et al., 2023). A main advantage of ML is the ability to parameterize general relationships from training data without predefined or explicit equations (Marzban, 2009). To date, a variety of ML models have been used for retrieval of various ocean parameters, including the diffuse attenuation coefficient (Chen et al., 2015), particle backscattering coefficient (Sauzède et al., 2016), Chl$a$ concentration (Chen et al., 2021; Hu et al., 2021; Kolluru and Tiwari, 2022; Mukonza and Chiang, 2022; Syariz et al., 2020), and reconstructions of ocean color data (Chen et al., 2019; Fasnacht et al., 2022; Krasnopolsky et al., 2016). The current study aims to (1) parameterize CSD models for the Pacific Arctic using spectral features of $R_{rs}(\lambda)$ and $a_{ph}(\lambda)$, (2) assess satellite algorithm performance using an *in situ* dataset, and (3) compare newly developed models with the previously developed PCA-based CSD model. The updated CSD model provides accurate estimates of spatiotemporal variations in PSC

in the Pacific Arctic, providing key information on how recent environmental changes are affecting the foundation of marine food webs in a changing Arctic.

Table 1. Definitions and units of all symbols used in the text, figures, and equations.

| Symbol | Definition | Unit |
|---|---|---|
| Chl$a$ | Chlorophyll-$a$ concentration | mg m$^{-3}$ |
| Chl$a_0$ | Chl$a$ at reference diameter $D_0$ | mg m$^{-3}$ |
| Chl$a_{total}$ | Total Chl$a$ | mg m$^{-3}$ |
| Chl$a_{size}$ | Size-fractionated Chl$a$ in within a size bin from $D_1$ to $D_2$ | mg m$^{-3}$ |
| Chl$a_{size\_obs}$ | *In situ* Chl$a_{size}$ | mg m$^{-3}$ |
| $\eta$ | Exponent of the CSD | - |
| $\eta_{obs}$ | *In situ* $\eta$ retrieved from in-situ Chl$a_{size\_obs}$ | - |
| $\eta_{MDLobs}$ | Estimated $\eta$ using the CSD model from *in situ* data | - |
| $\eta_{MDLsat}$ | Estimated $\eta$ using the CSD model from satellite data | - |
| $F_{size}$ | Fractional contribution of pico-, nano-, micro-plankton to Chl$a_{total}$ | - |
| $F_{size\_obs}$ | *In situ* $F_{size}$ retrieved from Chl$a_{total}$ and Chl$a_{size\_obs}$ | - |
| $F_{size\_MDL}$ | Estimated $F_{size}$ using the CSD model | - |
| $D_0$ | Reference diameter (0.7 μm) | μm |
| $D_{min}$ | Lower bound for size integration (0.7 μm) | μm |
| $D_{max}$ | Upper bound for size integration (200 μm) | μm |
| $D_1$ | Lower size limit of Chl$a_{size}$ | μm |
| $D_2$ | Upper size limit of Chl$a_{size}$ | μm |
| $\lambda$ | Wavelength | nm |
| $R_{rs}(\lambda)$ | Remote sensing reflectance at $\lambda$ | sr$^{-1}$ |
| $R_{rs\_obs}(\lambda)$ | *In situ* $R_{rs}(\lambda)$ | sr$^{-1}$ |
| $R_{rs\_sat}(\lambda)$ | Satellite $R_{rs}(\lambda)$ | sr$^{-1}$ |
| $\hat{R}_{rs\_obs}(\lambda)$ | *In situ* $R_{rs}(\lambda)$ normalized with Eq. (5) | - |
| $a_{ph}(\lambda)$ | Absorption coefficient of phytoplankton at $\lambda$ | m$^{-1}$ |
| $a_{ph\_obs}(\lambda)$ | *In situ* $a_{ph}(\lambda)$ | m$^{-1}$ |
| $a_{ph\_QAA}(\lambda)$ | $a_{ph}(\lambda)$ estimated using modified QAA | m$^{-1}$ |
| $a_{ph\_QAAobs}(\lambda)$ | Estimated $a_{ph\_QAA}(\lambda)$ from *in situ* $R_{rs}(\lambda)$ | m$^{-1}$ |
| $a_{ph\_QAAsat}(\lambda)$ | Estimated $a_{ph\_QAA}(\lambda)$ from satellite $R_{rs}(\lambda)$ | m$^{-1}$ |
| $\hat{a}_{ph\_obs}(\lambda)$ | *In situ* $R_{rs}(\lambda)$ normalized with Eq. (5) | - |
| $a_p(\lambda)$ | Absorption coefficient of particles at $\lambda$ | m$^{-1}$ |
| $a_{p\_obs}(\lambda)$ | *In situ* $a_p(\lambda)$ | m$^{-1}$ |
| $a_{NAP}(\lambda)$ | Absorption coefficient of NAP at $\lambda$ | m$^{-1}$ |
| $a_{NAP\_obs}(\lambda)$ | *In situ* $a_{NAP}(\lambda)$ | m$^{-1}$ |
| $a_{CDOM}(\lambda)$ | Absorption coefficient of CDOM at $\lambda$ | m$^{-1}$ |
| $a_{CDOM\_obs}(\lambda)$ | *In situ* $a_{CDOM}(\lambda)$ | m$^{-1}$ |
| $a_w(\lambda)$ | Absorption coefficient of pure water at $\lambda$ | m$^{-1}$ |
| $S_{dg}$ | Spectral slope of the absorption coefficient of combined CDOM and NAP | nm$^{-1}$ |
| $L_w(\lambda)$ | Water-leaving radiance at $\lambda$ | W m$^{-2}$ sr$^{-1}$ nm$^{-1}$ |
| $E_s(\lambda)$ | Downwelling irradiance above surface at $\lambda$ | W m$^{-2}$ nm$^{-1}$ |
| $\beta_0$ | Intercept in PCA-based CSD model | - |
| $C_j$ | Coefficients in PCA-based CSD model at wavelength $j$ | - |

## 2    Material and methods

An updated CSD model is proposed in this study to enable reasonable estimation of spatiotemporal variations in PSC for optically complex Pacific Arctic waters. See Section S1–S4 in Supplement for complete materials and methods.

### 2.1    *In situ* data

Multiple research cruises were conducted in the Pacific Arctic during the summer months from 2007 to 2021 (Table 2). A total of 177 open ocean and coastal sampling locations were visited in the sub-Arctic Bering Sea and the west Beaufort Sea, including the Stefansson Sound near Prudhoe Bay along the northern coast of Alaska (Figure 1). A companion map, color-coded by cruise year, is provided in Figure S1. At each station, spectral radiometric measurements were made during daylight hours, and water samples were collected for $a_{ph}(\lambda)$ and size-fractionated Chl$a$ (hereafter referred to as $a_{ph\_obs}(\lambda)$ and Chl$a_{size\_obs}$, respectively).

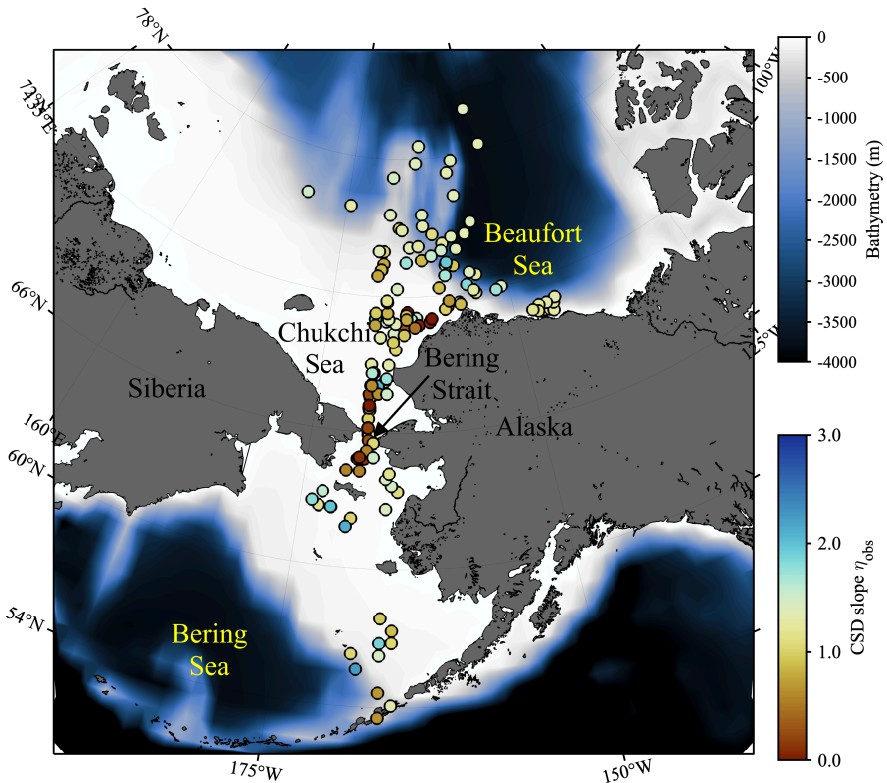

Figure 1. Sampling locations of *in situ* data used in this study. Colors of each plot indicate the exponent of chlorophyll-*a* (Chl*a*) size distribution (CSD slope; $\eta_{obs}$), whereas background color represent the bathymetry.

Table 2. Details of cruises, number of samples (*N*) obtained during each cruise, and filter pore sizes used to collect size fractionated chlorophyll-*a* samples. Note that the cruise period indicates the date span of *in situ* data collected.

| Cruise period (mm/dd/yyyy) | Cruise ID | Vessel | $N$ | Filter pose size |
|---|---|---|---|---|
| 07/25–08/14/2007 | OS180 | T/S Oshoro-maru | 20 | 20, 5, and 0.7 μm |
| 09/11–10/10/2009 | MR09-03 | R/V Mirai | 12 | 10, 5, and 0.7 μm |
| 09/04–10/13/2010 | MR10-05 | R/V Mirai | 28 | 10, 5, and 0.7 μm |
| 09/13–10/02/2012 | MR12-E03 | R/V Mirai | 12 | 20, 2, and 0.7 μm |
| 06/06–07/17/2013 | OS255 | T/S Oshoro-maru | 34 | 20, 2, and 0.7 μm |
| 08/31–10/04/2013 | MR13-06 | R/V Mirai | 32 | 20, 2, and 0.7 μm |
| 08/30–09/22/2016 | MR16-06 | R/V Mirai | 18 | 20, 2, and 0.7 μm |
| 07/09–07/21/2017 | OS040 | T/S Oshoro-maru | 11 | 20, 2, and 0.7 μm |
| 08/13–08/15/2021 | PB21 | R/V Ukpik | 10 | 20, 2, and 0.7 μm |

### 2.1.1 Phytoplankton pigments

Chl$a_{size\_obs}$ was determined using a 10-AU fluorometer (Turner Designs), except for ten samples from the 2021 cruise in Prudhoe Bay (PB21), for which Chl$a_{size\_obs}$ was determined using high performance liquid chromatography (HPLC). HPLC analysis provides the concentration of not only Chl$a$ but also other major phytoplankton pigments (i.e., fucoxanthin, peridinin, 19'-hexanoyloxyfucoxanthin, 19'-butanoylofucoxanthin, alloxanthin, chlorophyll-$b$, neoxanthin, prasinoxanthin, violaxanthin, lutein, and zeaxanthin). At each station in all the cruises, both fractionated and unfractionated (i.e., without filtration using filters of different pore sizes for size fractionation) samples were collected. Unfractionated HPLC samples were collected at each station in all the cruises.

### 2.1.2 Absorption coefficient

Particles in surface seawater samples were collected on a GF/F filter until the filter had sufficient coloration to measure $a_{ph\_obs}(\lambda)$. The absorption coefficient of particles ($a_{p\_obs}(\lambda)$) on the filter was measured in the spectral range from 300 to 850 nm at 1 nm intervals using an MPS-2400 (Shimadzu Corporation), MPS-2450 (Shimadzu Corporation) or Cary 100 (Agilent Technologies) spectrophotometer. The quantitative filter technique (QFT) was used to determine $a_{ph\_obs}(\lambda)$ for samples measured with the MPS-2400 and MPS-2450 instruments (i.e., all cruises but PB21), following the procedure described by Mitchell (Mitchell, 1990), whereas $a_{ph\_obs}(\lambda)$ for the PB21 samples was determined with GF/F filters placed inside a 15-cm integrating sphere connected to the Cary 100 (IOCCG, 2018). Following the measurement for $a_{p\_obs}(\lambda)$, the absorption coefficient of NAP ($a_{NAP\_obs}(\lambda)$) was measured after soaking the filter in 95% methanol or sodium hypochlorite, and $a_{ph\_obs}(\lambda)$ was finally obtained by subtracting $a_{NAP\_obs}(\lambda)$ from $a_{p\_obs}(\lambda)$. The absorption coefficient of CDOM ($a_{CDOM\_obs}(\lambda)$) at wavelengths from 250 to 750 nm at 1 nm intervals was measured using the same spectrophotometers as for the particulate absorption measurements, with the exception of the PB21 samples, which were analyzed using a Cary 300 (Agilent Technologies) spectrophotometer with 5-cm quartz cuvettes.

### 2.1.3 Remote sensing reflectance

*In situ* spectral radiance and irradiance measurements were acquired using a PRR-800/810 (Biospherical Instruments), C-OPS (Biospherical Instruments), or HyperPro (Satlantic) spectroradiometer. Each spectroradiometer has different spectral resolutions and ranges: the PRR-800/810 and C-OPS collected at 17 (380 to 765 nm) and 19 wavelengths (320 to 875 nm), respectively, whereas the HyperPro acquired data between 400 and 800 nm at approximately 3 nm intervals. Remote sensing reflectance ($R_{rs\_obs}(\lambda)$) was calculated as the ratio of the water-leaving radiance ($L_w(\lambda)$) to the above-water downward spectral irradiance ($E_s(\lambda)$):

$$R_{rs\_obs}(\lambda) = L_w(\lambda)/E_s(\lambda). \tag{1}$$

$R_{rs\_obs}(\lambda)$ was resampled at ten MODIS bands in the visible range (i.e., 412, 443, 469, 488, 531, 547, 555, 645, 667, and 678 nm) from the original wavelengths of each instrument using spline interpolation (Wang et al., 2015). Finally, a modified version of the Quasi-Analytical Algorithm (QAA; (Lee et al., 2002)) for the Pacific Arctic (Fujiwara et al., 2016) was used to estimate $a_{ph}(\lambda)$ ($a_{ph\_QAA}(\lambda)$) from *in situ* $R_{rs}(\lambda)$ ($R_{rs\_obs}(\lambda)$) and satellite $R_{rs}(\lambda)$ ($R_{rs\_sat}(\lambda)$). Here, $a_{ph\_QAA}(\lambda)$ estimated from $R_{rs\_obs}(\lambda)$ and $R_{rs\_sat}(\lambda)$ is denoted as $a_{ph\_QAAobs}(\lambda)$ and $a_{ph\_QAAsat}(\lambda)$, respectively. To avoid the retrieval of negative $a_{ph\_QAA}(\lambda)$, the modified version of QAA uses an optimized spectral slope of the absorption coefficient of combined CDOM and NAP ($S_{dg}$) obtained by reconstructing the $S_{dg}$ based on a dataset collected in the Pacific Arctic (Fujiwara et al., 2016). The $a_{ph\_QAAobs}(\lambda)$ was used to validate the performance of the modified version of the QAA by comparing it with $a_{ph\_obs}(\lambda)$.

### 2.1.4 Pigment-based identification of phytoplankton taxonomic composition

An open-source R software package, *phytoclass* (ver 1.0.0), was used to determine the Chl*a* biomass of different phytoplankton groups from their accessory pigments (Hayward et al., 2023). The *phytoclass* package is a Chl*a* taxonomic partitioning software package similar to the widely used CHEMTAX software (Mackey et al., 1996). However, *phytoclass* has been shown to be more accurate and does not rely on initial assumptions of pigment to Chl*a* ratios for each phytoplankton group (Hayward et al., 2023). Eight target taxonomic groups (diatoms, chrysophytes, dinoflagellates, prymnesiophytes, chlorophytes, prasinophytes, cryptophytes, and cyanobacteria) and 11 marker pigments for each taxonomic group (peridinin, 19'-butanoyloxyfucoxanthin, fucoxanthin, 19'-hexanoyloxyfucoxanthin, neoxanthin, prasinoxanthin, violaxanthin, alloxanthin, lutein, zeaxanthin, and chlorophyll-*b*) were selected following (Zhuang et al., 2016), as these groupings have been used previously for CHEMTAX analysis in the Chukchi Sea shelf region.

## 2.2 Satellite data

The MODIS sensor onboard NASA's Aqua satellite (MODIS-A), operational since 2002, provides the longest time series among all currently operational ocean color sensors, which is an attractive advantage for decadal-scale monitoring and retrospective analyses. Level-3 standard mapped images of 4 km spatial resolution monthly climatological $R_{rs\_sat}(\lambda)$ at ten

bands in the visible range (i.e., 412, 443, 469, 488, 531, 547, 555, 645, 667, and 678 nm) and daytime sea surface temperature (SST) derived by MODIS-A (version R2022.0) were downloaded from NASA's Ocean Color website. The $R_{rs\_sat}(\lambda)$ data were then used to compute $a_{ph\_QAAsat}(\lambda)$ by using the modified QAA algorithm (Fujiwara et al., 2016).

## 2.3 Chlorophyll-*a* size distribution model

The exponent of the CSD ($\eta$), representing the size structure of phytoplankton communities, was determined following the method of Waga et al. (2017). Assuming the CSD follows a Junge-type power law distribution, the total Chl*a* (Chl*a*$_{total}$) and Chl*a*$_{size}$ in a size range from $D_1$ to $D_2$ can be expressed as follows:

$$\mathrm{Chl}a_{total} = \int_{D_{min}}^{D_{max}} \mathrm{Chl}a_0 \left(\frac{D}{D_0}\right)^{-\eta} dD, \tag{2}$$

$$\mathrm{Chl}a_{size} = \int_{D_1}^{D_2} \mathrm{Chl}a_0 \left(\frac{D}{D_0}\right)^{-\eta} dD, \tag{3}$$

where Chl*a*$_0$ is the Chl*a* at a reference diameter $D_0$ (here, 0.7 μm). In this study, $D_{min}$ and $D_{max}$ were defined as 0.7 μm and 200 μm, respectively. $\eta$ was derived as the slope of the linear regression in log-space computations between the inverse log-transformed median diameters (from $D_1$ to $D_2$), and Chl*a*$_{size}$ normalized by the bin width. An advantage of the CSD model is its robustness when using different sets of Chl*a*$_{size}$ to retrieve $\eta$ (Waga et al., 2017).

A large $\eta$ indicates a greater contribution of smaller-sized phytoplankton, whereas a small $\eta$ suggests that larger-sized phytoplankton dominate. The fraction of Chl*a*$_{size}$ can be derived using $\eta$ as follows:

$$F_{size} = \frac{\mathrm{Chl}a_{size}}{\mathrm{Chl}a_{total}} = \frac{\int_{D_1}^{D_2} \mathrm{Chl}a_0 \left(\frac{D}{D_0}\right)^{-\eta} dD}{\int_{D_{min}}^{D_{max}} \mathrm{Chl}a_0 \left(\frac{D}{D_0}\right)^{-\eta} dD} = \frac{D_2^{1-\eta} - D_1^{1-\eta}}{200^{1-\eta} - 0.7^{1-\eta}}. \tag{4}$$

In this study, the size ranges for pico-, nano-, and micro-phytoplankton were defined as 0.7–2 μm, 2–20 μm, and 20–200 μm, respectively. To estimate the fraction of Chl*a* within the size ranges for pico- ($F_{pico}$), nano- ($F_{nano}$), and micro-phytoplankton ($F_{micro}$), $D_1$ and $D_2$ in Eq. (4) were set as the lower and upper limits of each size range. For clarification purposes, the size fractions determined from *in situ* Chl*a*$_{size}$ observations are denoted as $F_{size\_obs}$, whereas those estimated through a CSD model with Eq. (4) using $\eta$ were represented as $F_{size\_MDL}$.

## 2.4 Model development

The CSD model was trained using 70% of the entire dataset (i.e., training subset), randomly determined using the MATLAB *randsample* function (R2025b), while the remaining 30% was used for final validation (i.e., validation subset). The details of model development based on the PCA and supervised ML approaches are described in sections 2.4.1 and 2.4.2, respectively.

### 2.4.1 PCA approach

The previous version of the CSD model for the Pacific Arctic (Waga et al., 2019a) used the spectral shape of $a_{ph}(\lambda)$ to estimate $\eta$. To capture the spectral features of $a_{ph}(\lambda)$, PCA was applied to normalized $a_{ph\_obs}(\lambda)$ ($\hat{a}_{ph\_obs}(\lambda)$) at ten MODIS-A bands. The formula for $\hat{a}_{ph\_obs}(\lambda)$ is:

$$\hat{a}_{ph\_obs}(\lambda) = [a_{ph\_obs}(\lambda) - \text{mean}(a_{ph\_obs}(\lambda))]/\text{std}(a_{ph\_obs}(\lambda)), \tag{5}$$

where $\text{mean}(a_{ph\_obs}(\lambda))$ and $\text{std}(a_{ph\_obs}(\lambda))$ are the spectral arithmetic mean and standard deviation of individual $a_{ph\_obs}(\lambda)$ spectra, respectively. The input values for the PCA comprise a matrix ($m \times N$) composed of $\hat{a}_{ph\_obs}(\lambda)$ values, where $m$ and $N$ are the number of the wavelengths and number of samples, respectively. Assuming the resulting PC scores correlate with $\eta$, $\eta$ was estimated as follows:

$$\eta = \left[\beta_0 + \exp\sum_{i=1}^{k}\beta_i S_i\right]^{-1}, \tag{6}$$

$$S_i = \sum_{j=1}^{m} w_{i,j}\, \hat{a}_{ph\_obs}(\lambda_j), \tag{7}$$

where $S_i$ and $w_{i,j}$ are the $i$th PC score and the loading factors for $i$th PC at wavelength $j$. In addition, $m$ and $k$ represent the number of wavelengths and the number of PCs ($k = 4$ in this study). The model parameters $\beta_0$ and $\beta_i$ are the regression coefficients between $\eta$ and PC scores.

By substituting for the calculation of $S_i$ in Eq. (6), we obtained new equations as follows:

$$\eta = \left[\beta_0 + \exp\sum_{j=1}^{m} C_j \hat{a}_{ph\_obs}(\lambda_j)\right]^{-1}, \tag{8}$$

$$C_j = \sum_{i=1}^{k}\beta_i w_{i,j}, \tag{9}$$

where $\beta_0$ and $C_j$ are the final model parameters. Once the model parameters were determined based on $a_{ph\_obs}(\lambda)$, the same coefficients were used in the case of $a_{ph\_QAAobs}(\lambda)$ and $a_{ph\_QAAsat}(\lambda)$ to produce estimates of $\eta$. For the $R_{rs}$-based models,

normalized $R_{rs\_obs}(\lambda)$ ($\hat{R}_{rs\_obs}(\lambda)$) was calculated in the same manner as Eq. (5), and $\eta$ was determined by employing $\hat{R}_{rs\_obs}(\lambda)$ in Eqs. (6)–(9) in place of $\hat{a}_{ph\_obs}(\lambda)$. Note that $\eta$ determined by Chl$a_{size\_obs}$, estimated through the CSD model using *in situ* measurements ($\hat{a}_{ph\_obs}(\lambda)$ or $\hat{R}_{rs\_obs}(\lambda)$) and satellite products ($\hat{a}_{ph\_QAAsat}(\lambda)$ or $\hat{R}_{rs\_sat}(\lambda)$) are denoted as $\eta_{obs}$, and $\eta_{MDLobs}$ and $\eta_{MDLsat}$, respectively.

### 2.4.2 Supervised ML approach

In addition to the PCA approach used in prior work (Waga et al., 2017, 2019a, b, 2021a), CSD models were trained with various ML approaches. Since we know both the input (i.e., $\hat{R}_{rs\_obs}(\lambda)$ or $\hat{a}_{ph\_obs}(\lambda)$) and corresponding output (i.e., $\eta_{obs}$) values, supervised ML was used to train CSD models. To this end, we leveraged the Regression Learner App in the MATLAB Statistics and Machine Learning toolbox, a user-friendly resource that enables simple data exploration, feature selection, specification of validation schemes, model training, and model evaluation. This application includes commonly used regression methods, e.g., linear regression models, regression trees, Gaussian process regression models, support vector machines, kernel approximation models, ensembles of regression trees, and neural network regression models.

To avoid the possibility of missing certain representative samples and/or overfitting the models, repeated five-fold cross-validation (ten repeats) was carried out by randomly dividing the training subset into five equally sized sets (or five-folds). Evaluation of the trained models was performed five times, each time excluding one-fold from the training subset and using it for validation. Each observation in the training subset was assigned to an individual group and stayed in that group for the duration of the procedure so that each observation was allowed to be used one time for testing and four times for training the model. Finally, the performance of the trained models was determined as the average of the performance metrics from the five iterations.

The MATLAB Regression Learner App returns three other statistical metrics besides the coefficient of determination ($r^2$): the root mean square error (RMSE), mean squared error (MSE), and mean absolute error (MAE) between the observed and predicted values, defined as:

$$\text{RMSE} = \sqrt{\sum_{n=1}^{N} (X_n - Y_n)^2 \Big/ N}, \tag{10}$$

$$\text{MSE} = \sum_{n=1}^{N} (X_n - Y_n)^2 \Big/ N, \tag{11}$$

$$\text{MAE} = \sum_{n=1}^{N} |X_n - Y_n| \Big/ N, \tag{12}$$

where $X_n$ and $Y_n$ represent the $n^{th}$ observed and predicted values, respectively. Once CSD models based on each ML method were finalized, the best ML-based CSD model for each predictor (i.e., $\hat{R}_{rs\_obs}(\lambda)$ and $\hat{a}_{ph\_obs}(\lambda)$) was determined based on the four aforementioned statistical metrics. Once the best-performing models for $\hat{R}_{rs\_obs}(\lambda)$ and $\hat{a}_{ph\_obs}(\lambda)$ among diverse regression methods were determined, they were used for final validation and further analysis.

## 2.5    Model validation metrics

The performance of the resulting PCA-based CSD models and the best-performing ML-based CSD models were compared using the validation subset. Bias is a key metric for the performance assessment of satellite products (Seegers et al., 2018), defined as:

$$Bias = 10^\wedge \left( \sum_{n=1}^{N} (X_n - Y_n) \Big/ N \right) \tag{13}$$

Following recommended validation procedures for satellite ocean color algorithms (Seegers et al., 2018), the performance of the CSD models, as well as the modified QAA, was evaluated based on MAE (Eq. (12)) and bias.

## 3    Results

### 3.1    Phytoplankton size structure and taxonomic composition

The measured exponent of CSD ($\eta_{obs}$) values ranged from 0 to 2.24 with corresponding Chl$a_{total\_obs}$ values of 18.84 and 0.05 mg m$^{-3}$, respectively (Table 3). Figure 2 depicts the Chl$a_{total\_obs}$ and $\eta_{obs}$ values with regard to the relative contributions of $F_{size\_obs}$. High Chl$a_{total\_obs}$ was characterized by communities having a predominant contribution of $F_{micro\_obs}$ and correspondingly lower contributions of both $F_{pico\_obs}$ and $F_{nano\_obs}$. A similar but opposite pattern was found in $\eta_{obs}$, with small $\eta_{obs}$ values clearly associated with large $F_{micro\_obs}$. This opposite pattern resulted from the fact that small $\eta_{obs}$ values represent significant contributions of $F_{micro\_obs}$ essentially associated with high Chl$a_{total\_obs}$. In addition, $F_{micro\_obs}$ and $F_{pico\_obs}$ ranged between 0.01–0.94 and 0.00–0.80, respectively, suggesting that our dataset covered a wide range of PSCs in the Pacific Arctic. According to Eq. (4), the smallest $\eta_{obs}$ corresponded to 0.9, 0.09, and 0.01 of $F_{size\_MDL}$ for micro-, nano-, and pico-phytoplankton, whereas the largest $\eta_{obs}$ corresponded to 0.02, 0.26, and 0.73, respectively.

Table 3. Summary statistics of primary variables used in this study. Note that these variables were determined by *in situ* observations. Abbreviation: Chl*a*, chlorophyll-*a*; $\eta$, exponent of Chl*a* size distribution (CSD); $F_{size}$, fractional contribution of micro-, nano-, and pico-plankton; $a_{ph}(443)$ phytoplankton absorption coefficient at 443 nm; $a_{NAP}(443)$, absorption coefficient of non-algal particles (NAP) at 443 nm; $a_{CDOM}(443)$, absorption coefficient of colored dissolved organic matter (CDOM) at 443 nm; and $R_{rs}(443)$, remote sensing reflectance at 443 nm.

| | $Chla_{total\_obs}$ (mg m$^{-3}$) | $\eta_{obs}$ | $F_{micro\_obs}$ | $F_{nano\_obs}$ | $F_{pico\_obs}$ | $a_{ph\_obs}(443)$ (m$^{-1}$) | $a_{NAP\_obs}(443)$ (m$^{-1}$) | $a_{CDOM\_obs}(443)$ (m$^{-1}$) | $R_{rs\_obs}(443)$ ($\times10^2$ sr$^{-1}$) |
|---|---|---|---|---|---|---|---|---|---|
| Mean | 0.54 | 1.02 | 0.36 | 0.32 | 0.32 | 0.04 | 0.03 | 0.09 | 0.30 |
| Median | 0.40 | 1.08 | 0.35 | 0.32 | 0.30 | 0.02 | 0.01 | 0.06 | 0.30 |
| Std | 3.62 | 0.50 | 0.27 | 0.11 | 0.20 | 0.05 | 0.11 | 0.08 | 0.11 |
| Min | 0.05 | 0.00 | 0.01 | 0.02 | 0.00 | 0.00 | 0.00 | 0.01 | 0.05 |
| Max | 18.84 | 2.24 | 0.94 | 0.51 | 0.80 | 0.32 | 1.18 | 0.40 | 0.66 |

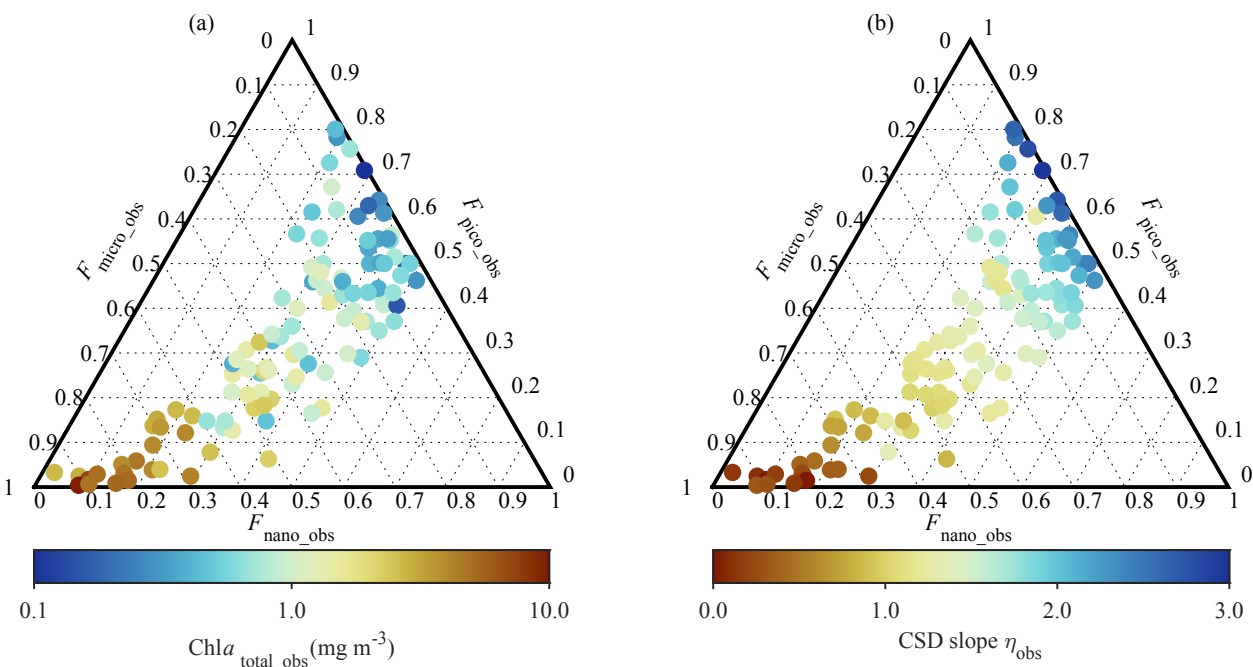

Figure 2. Ternary diagrams depicting phytoplankton size composition. Each diagram illustrates fractional contribution of micro- ($F_{micro\_obs}$), nano- ($F_{micro\_obs}$), and picophytoplankton ($F_{micro\_obs}$) to total phytoplankton biomass, colored with (a) total Chl*a* (Chl*a*$_{total\_obs}$) and (b) $\eta_{obs}$, respectively.

Figure 3 illustrates the biomass and fractional contribution to total Chl$a$ of phytoplankton taxa determined by *phytoclass*, with respect to $\eta_{obs}$. The pigment ratios used in this study are detailed in Table S1. Diatoms dominated in terms of both biomass and fractional contribution for small $\eta_{obs}$ values and gradually decreased as the $\eta_{obs}$ value increased ($p < 0.01$). A similar but opposite pattern was observed for prymnesiophytes, indicating a gradual increase in the fractional contribution with increasing $\eta_{obs}$ values ($p < 0.01$). Interestingly, diatoms and prymnesiophytes were the only taxa that dominated the phytoplankton communities, while other taxa remained only minor contributors across the $\eta_{obs}$ range. More specifically, prasinophytes and cryptophytes showed slight increases in their fractional contribution up to >0.30 at $\eta_{obs}$ values ranging from 0.70–2.00, while their Chl$a$ biomass in all cases remained less than 0.20 mg m$^{-3}$. Other taxa showed negligible variations in biomass, whereas their fractional contributions fluctuated in response to reduced Chl$a$ for the entire phytoplankton community but was statistically insignificant ($p \geq 0.01$).

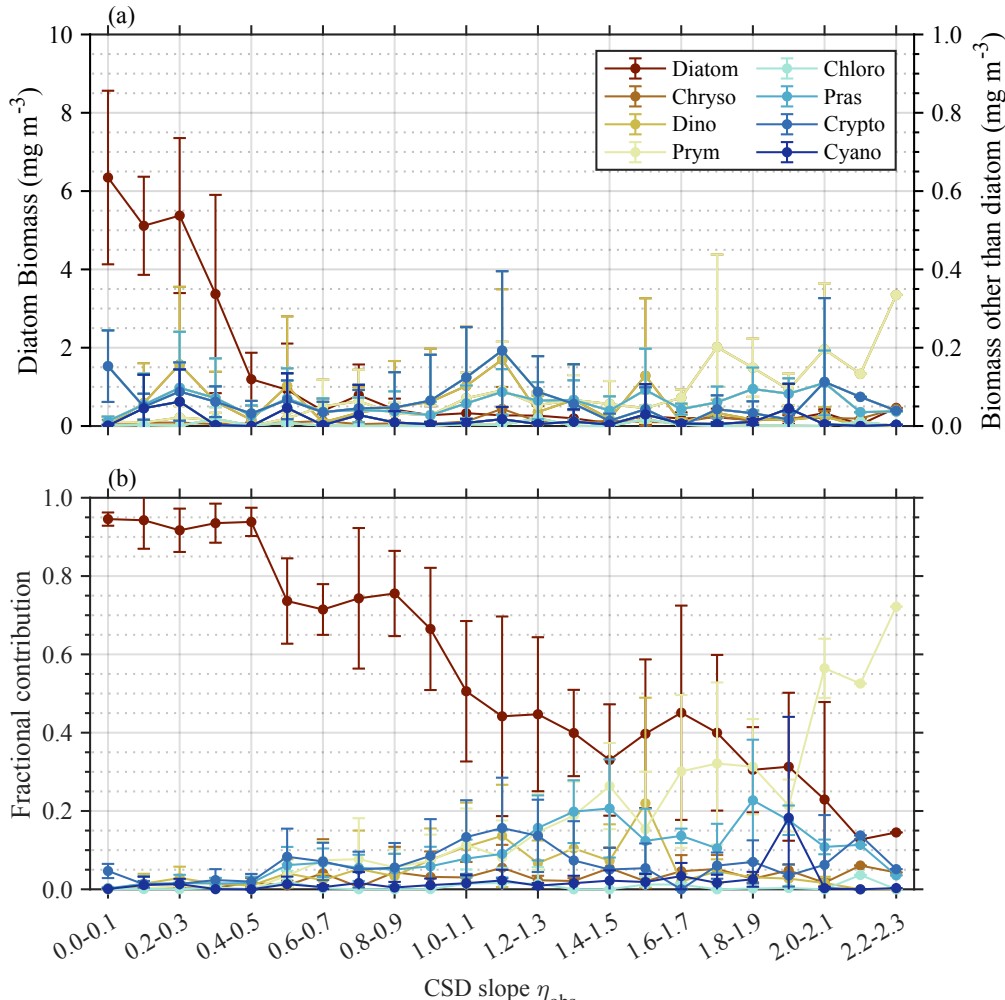

Figure 3. Variations in major phytoplankton groups with reference to CSD slope. (a) Biomass and (b) fractional contribution of each phytoplankton taxa to total phytoplankton biomass (Chl*a*) determined by *phytoclass*, with respect to $\eta_{obs}$ value. Plots and vertical bars denote the average and standard deviations of each value within the respective $\eta_{obs}$ bins. Abbreviations: Chryso, chrysophytes; Dino, dinoflagellates; Prym, prymnesiophytes; Chloro, chlorophytes; Pras, prasinophytes; Crypto, cryptophytes; Cyano, cyanobacteria.

## 3.2    Phytoplankton absorption and remote sensing reflectance spectra

Since the Pacific Arctic is characterized as optically complex, i.e., the contributions of different water constituents (phytoplankton, NAP, and CDOM) are highly variable, the fractional contribution of each constituent to the total absorption by seawater ($a_{total\_obs}(\lambda)$) was investigated using *in situ* data (Figure S4). The ratio of $a_{ph\_obs}(\lambda)$ to $a_{total\_obs}(\lambda)$ was typically <0.30, even at wavelengths of maximum pigment absorption (i.e., 443, 469, and 488 nm) and weak pure water absorption ($a_w(\lambda)$), whereas $a_{CDOM\_obs}(\lambda)$ comprised $0.66 \pm 0.15$ (mean ± std) of $a_{total\_obs}(412)$. At longer wavelengths (i.e., 645, 667, and 678 nm), $a_w(\lambda)$ contributed significantly to total absorption, with average values of >0.95. Overall, phytoplankton was the dominant constituent to $a_{total\_obs}(443)$ for only 30 of the 177 samples, suggesting that estimations of $a_{ph}(\lambda)$ from $R_{rs}(\lambda)$ using the QAA algorithm are likely to have large uncertainties for the majority of samples due to the significant contributions to absorption by other water constituents.

Figure 4 shows spectral variations in $R_{rs\_obs}(\lambda)$, $a_{ph\_obs}(\lambda)$, $\hat{R}_{rs\_obs}(\lambda)$, and $\hat{a}_{ph\_obs}(\lambda)$ at ten MODIS-A bands, with respect to $\eta_{obs}$. Larger spectral variations in $R_{rs\_obs}(\lambda)$, with a distinct peak at green wavelengths (i.e., 531, 547, and 555 nm), were found for smaller $\eta_{obs}$ values, whereas larger $\eta_{obs}$ values corresponded to relatively flat spectral shapes, with only small peaks at shorter wavelengths (i.e., 469 and 488 nm). $a_{ph\_obs}(\lambda)$ also showed similar differences in spectral shape and magnitude with $\eta_{obs}$ values, except with peaks at blue wavelengths. In contrast, $\hat{R}_{rs\_obs}(\lambda)$ and $\hat{a}_{ph\_obs}(\lambda)$ emphasize only spectral shape by normalizing the range of variability in $R_{rs\_obs}(\lambda)$ and $a_{ph\_obs}(\lambda)$ (Figure 6c, d). Regarding $\hat{a}_{ph\_obs}(\lambda)$, sharper peaks at blue wavelengths (i.e., 412, 443, and 469 nm) with the maximum value at 443 nm were observed for large $\eta_{obs}$. Moreover, $\hat{a}_{ph\_obs}(\lambda)$ increased more prominently with increasing wavelength from its minimum near 550 nm at smaller $\eta_{obs}$, whereas larger $\eta_{obs}$ corresponded to less pronounced increases in $\hat{a}_{ph\_obs}(\lambda)$ over this spectral range. Overall, the spectral features of $\hat{R}_{rs\_obs}(\lambda)$ and $\hat{a}_{ph\_obs}(\lambda)$ exhibited clear variations associated with $\eta_{obs}$ values, with $\hat{R}_{rs\_obs}(\lambda)$ exhibiting larger variations associated with $\eta_{obs}$ across the wide range of wavelengths compared to $\hat{a}_{ph\_obs}(\lambda)$. $\hat{a}_{ph\_obs}(\lambda)$ also exhibited larger spectral variations, but differences associated with $\eta_{obs}$ were smaller in magnitude. The performance of the modified QAA for MODIS-A bands, determined by comparing $a_{ph\_QAAobs}(\lambda)$ with $a_{ph\_obs}(\lambda)$, is shown in Table S2. According to the validation results, $a_{ph}(\lambda)$ values at longer wavelengths (645, 667, and 678 nm) exhibited poor QAA estimation accuracy and were removed from the model development based on PCA and ML approaches. It is noteworthy that the MAE for these wavelengths represents between 25% and 30% of the pure water values (Pope and Fry, 1997). While this might appear large in an absolute sense, the red portion of the spectrum contains limited phytoplankton taxonomic information outside of the chlorophyll absorption band at 678 nm (Huot et al., 2005).

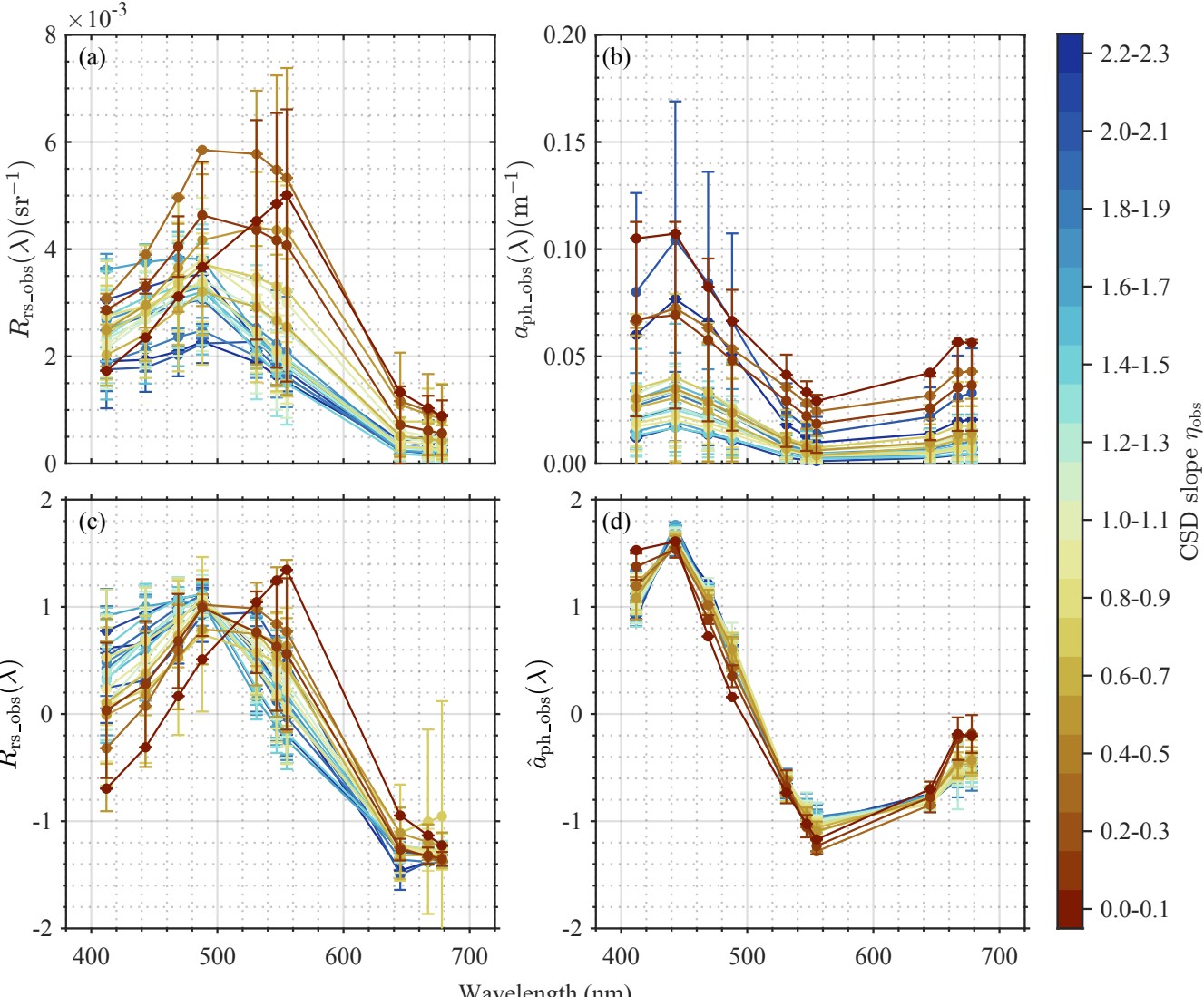

Figure 4. Spectral variations in key optical properties. Spectral variations in (a) remote sensing reflectance ($R_{rs\_obs}(\lambda)$), (b) $a_{ph\_obs}(\lambda)$, (c) normalized $R_{rs\_obs}(\lambda)$ ($\hat{R}_{rs\_obs}(\lambda)$), and (d) normalized $a_{ph\_obs}(\lambda)$ ($\hat{a}_{ph\_obs}(\lambda)$) with respect to $\eta_{obs}$. Vertical bars represent the standard deviations at each wavelength for each $\eta_{obs}$ range.

### 3.4 CSD model development: PCA approach

The spectral features of $\hat{R}_{\mathrm{rs\_obs}}(\lambda)$ and $\hat{a}_{\mathrm{ph\_obs}}(\lambda)$ captured by PCA were used to develop the CSD model. Variations in the loading factors, which describe how much each variable contributes to a particular principal component at ten wavelengths (i.e., 412, 443, 469, 488, 531, 547, 555, 645, 667, and 678 nm) and seven MODIS-A bands (i.e., 412, 443, 469, 488, 531, 547, and 555 nm) for $\hat{R}_{\mathrm{rs\_obs}}(\lambda)$ and $\hat{a}_{\mathrm{ph\_obs}}(\lambda)$, respectively, are shown in Figures S2 and S3.

The spectral features captured by PCA demonstrate optical signatures of $\hat{R}_{\mathrm{rs\_obs}}(\lambda)$ and $\hat{a}_{\mathrm{ph\_obs}}(\lambda)$. The regression coefficients $\beta_0$ and $\beta_i$ of the logistic-type function (Eqs. (8) and (9)) were therefore determined by least squares regression between the first four PC scores of $\hat{R}_{\mathrm{rs\_obs}}(\lambda)$ or $\hat{a}_{\mathrm{ph\_obs}}(\lambda)$ and $\eta_{\mathrm{obs}}$. The resulting regression coefficients were then used to compute the model parameter $C_j$ (Eq. (8)). Here, PCA and subsequent procedures for $\beta_i$ and $C_j$ retrievals were conducted separately for two sample groups exhibiting either $a_{\mathrm{ph}}(412) \geq a_{\mathrm{ph}}(469)$ or $a_{\mathrm{ph}}(412) < a_{\mathrm{ph}}(469)$ regarding $\hat{a}_{\mathrm{ph\_obs}}(\lambda)$, whereas the procedures for $\hat{R}_{\mathrm{rs\_obs}}(\lambda)$ were performed on the entire dataset (unpartitioned) for model training. The partitioning of the model parameters for $\hat{a}_{\mathrm{ph\_obs}}(\lambda)$ was based on the trial-and-error approach (Waga et al., 2017) because a single combination of regression coefficients cannot capture the entire variations in the spectral shape of $\hat{a}_{\mathrm{ph\_obs}}(\lambda)$ in response to changing $\eta_{\mathrm{obs}}$. The partitioning sequence aimed to avoid underestimation that was observed for higher $\eta_{\mathrm{obs}}$ (Waga et al., 2017). Since no specific pattern in $\eta_{\mathrm{obs}}$ estimation was identified for $\hat{R}_{\mathrm{rs\_obs}}(\lambda)$, this study did not exploit the portioning approach for $\hat{R}_{\mathrm{rs\_obs}}(\lambda)$. The resulting model parameters are summarized in Table S3. The resulting PCA-based CSD models for $\hat{R}_{\mathrm{rs\_obs}}(\lambda)$ and $\hat{a}_{\mathrm{ph\_obs}}(\lambda)$ were hereafter denoted as CSD model$_{\mathrm{PCA}-\hat{R}_{\mathrm{rs}}(\lambda)}$ and CSD model$_{\mathrm{PCA}-\hat{a}_{\mathrm{ph}}(\lambda)}$, respectively.

### 3.5 CSD model development: supervised ML approach

Additional CSD models were developed using a supervised ML approach through MATLAB's Regression Learner App, setting $\hat{R}_{\mathrm{rs\_obs}}(\lambda)$ or $\hat{a}_{\mathrm{ph\_obs}}(\lambda)$ as input and $\eta_{\mathrm{obs}}$ as output. Performance statistics for the top five and bottom five models are presented in Table 4. Comprehensive results for the 28 models appear in Tables S4 ($\hat{R}_{\mathrm{rs\_obs}}(\lambda)$) and S5 ($\hat{a}_{\mathrm{ph\_obs}}(\lambda)$). The best model for $\hat{R}_{\mathrm{rs\_obs}}(\lambda)$ was a linear regression with linear preset, whereas that for $\hat{a}_{\mathrm{ph\_obs}}(\lambda)$ was a support vector machine (SVM) with medium Gaussian preset. These models achieved the best performance on the majority of four statistical metrics (i.e., RMSE, MSE, $r^2$, and MAE) relative to the other candidates and were thus selected as the ML-based CSD models for $\hat{R}_{\mathrm{rs\_obs}}(\lambda)$ and $\hat{a}_{\mathrm{ph\_obs}}(\lambda)$; hereafter, CSD model$_{\mathrm{LR}-\hat{R}_{\mathrm{rs}}(\lambda)}$ and CSD model$_{\mathrm{SVM}-\hat{a}_{\mathrm{ph}}(\lambda)}$, respectively. The model parameters for CSD model$_{\mathrm{LR}-\hat{R}_{\mathrm{rs}}(\lambda)}$ is reported in Table S6.

Upon statistical evaluation, we found random patterns in relationships between model performance and regression methods. For example, the linear regression with linear interaction preset showed the second worst performance while the standard linear preset showed the best performance among all 28 models tested with $\hat{R}_{\mathrm{rs\_obs}}(\lambda)$ as input. The SVM showed the best (medium

Gaussian preset) and worst (cubic preset) performance for $\hat{a}_{\text{ph\_obs}}(\lambda)$. The models trained with the neural network method tended to show poor estimation accuracy for both $\hat{R}_{\text{rs\_obs}}(\lambda)$ and $\hat{a}_{\text{ph\_obs}}(\lambda)$. Overall, the performance of the CSD models developed by the supervised ML approach varied largely among the regression methods used in the training process, indicating that care should be taken when choosing a regression method for model development.

Table 4. Training results of the top five and bottom five CSD models based on diverse machine learning approaches (i.e., model type and preset). The four statistical metrics, including the root mean square error (RMSE), mean squared error (MSE), coefficient of determination ($r^2$), and mean absolute error (MAE), are given as mean ± std derived from ten repeats of five-fold cross-validation.

| Predictor | Rank | Model type | Preset | RMSE | | | MSE | | | $r^2$ | | | MAE | | |
|---|---|---|---|---|---|---|---|---|---|---|---|---|---|---|---|
| $\hat{R}_{rs}(\lambda)$ | 1 | Linear Regression | Linear | 0.16 | ± | 0.01 | 0.03 | ± | 0.00 | 0.76 | ± | 0.02 | 0.12 | ± | 0.01 |
| | 2 | Linear Regression | Robust Linear | 0.16 | ± | 0.01 | 0.03 | ± | 0.00 | 0.76 | ± | 0.02 | 0.12 | ± | 0.00 |
| | 3 | SVM | Linear SVM | 0.17 | ± | 0.01 | 0.03 | ± | 0.00 | 0.74 | ± | 0.03 | 0.13 | ± | 0.00 |
| | 4 | Stepwise Linear Regression | Stepwise Linear | 0.18 | ± | 0.02 | 0.03 | ± | 0.01 | 0.70 | ± | 0.08 | 0.13 | ± | 0.01 |
| | 5 | Efficient Linear | Efficient Linear SVM | 0.18 | ± | 0.00 | 0.03 | ± | 0.00 | 0.69 | ± | 0.01 | 0.14 | ± | 0.00 |
| | ⋮ | | | | | | | | | | | | | | |
| | 24 | Neural Network | Medium Neural Network | 0.64 | ± | 0.21 | 0.45 | ± | 0.31 | -3.09 | ± | 2.87 | 0.27 | ± | 0.03 |
| | 25 | SVM | Quadratic SVM | 0.71 | ± | 0.41 | 0.66 | ± | 0.93 | -4.93 | ± | 8.37 | 0.25 | ± | 0.07 |
| | 26 | Neural Network | Wide Neural Network | 0.84 | ± | 0.36 | 0.82 | ± | 0.65 | -6.35 | ± | 5.88 | 0.32 | ± | 0.05 |
| | 27 | Linear Regression | Interactions Linear | 3.21 | ± | 1.26 | 11.74 | ± | 9.11 | -104.16 | ± | 80.99 | 0.55 | ± | 0.13 |
| | 28 | SVM | Cubic SVM | 24.10 | ± | 32.60 | 1537.37 | ± | 3416.27 | -13771.77 | ± | 30640.37 | 2.69 | ± | 3.08 |
| $\hat{a}_{ph}(\lambda)$ | 1 | SVM | Medium Gaussian SVM | 0.13 | ± | 0.01 | 0.02 | ± | 0.00 | 0.80 | ± | 0.02 | 0.10 | ± | 0.00 |
| | 2 | Gaussian Process Regression | Squared Exponential GPR | 0.13 | ± | 0.00 | 0.02 | ± | 0.00 | 0.80 | ± | 0.02 | 0.10 | ± | 0.00 |
| | 3 | Gaussian Process Regression | Matern 5/2 GPR | 0.13 | ± | 0.01 | 0.02 | ± | 0.00 | 0.80 | ± | 0.02 | 0.10 | ± | 0.00 |
| | 4 | Gaussian Process Regression | Rational Quadratic GPR | 0.13 | ± | 0.01 | 0.02 | ± | 0.00 | 0.79 | ± | 0.02 | 0.11 | ± | 0.00 |
| | 5 | Gaussian Process Regression | Exponential GPR | 0.14 | ± | 0.00 | 0.02 | ± | 0.00 | 0.78 | ± | 0.01 | 0.11 | ± | 0.00 |
| | ⋮ | | | | | | | | | | | | | | |
| | 24 | Neural Network | Bi-layered Neural Network | 0.57 | ± | 0.28 | 0.40 | ± | 0.38 | -3.65 | ± | 4.49 | 0.26 | ± | 0.03 |
| | 25 | Stepwise Linear Regression | Stepwise Linear | 0.71 | ± | 0.26 | 0.56 | ± | 0.35 | -5.51 | ± | 4.06 | 0.21 | ± | 0.03 |
| | 26 | Linear Regression | Interactions Linear | 1.18 | ± | 0.36 | 1.50 | ± | 0.96 | -16.46 | ± | 11.04 | 0.27 | ± | 0.04 |
| | 27 | SVM | Quadratic SVM | 1.35 | ± | 0.28 | 1.91 | ± | 0.76 | -21.18 | ± | 8.92 | 0.28 | ± | 0.03 |
| | 28 | SVM | Cubic SVM | 5.10 | ± | 2.66 | 32.39 | ± | 34.40 | -376.09 | ± | 402.63 | 0.67 | ± | 0.24 |

### 3.6    CSD model validation

Validation results of the four CSD models, i.e., CSD model$_{PCA-\hat{R}_{rs}(\lambda)}$, CSD model$_{PCA-\hat{a}_{ph}(\lambda)}$, CSD model$_{LR-\hat{R}_{rs}(\lambda)}$, and CSD model$_{SVM-\hat{a}_{ph}(\lambda)}$ are shown in Figure 5, with respect to the fractional contribution of $a_{ph\_obs}(443)$ to $a_{total\_obs}(443)$. The $\hat{a}_{ph\_obs}(\lambda)$-based models performed relatively well for both PCA and ML approaches, whereas, the PCA-based $\hat{R}_{rs\_obs}(\lambda)$ model underestimated $\eta_{obs}$, with the range of estimated values (~0.4–1.3) much lower than the measured range (~0.2–2.2). In addition, the ML-based models (CSD model$_{LR-\hat{R}_{rs}(\lambda)}$ and CSD model$_{SVM-\hat{a}_{ph}(\lambda)}$) showed better performance compared to the PCA-based models (CSD model$_{PCA-\hat{R}_{rs}(\lambda)}$ and CSD model$_{PCA-\hat{a}_{ph}(\lambda)}$). Overall, the CSD model$_{SVM-\hat{a}_{ph}(\lambda)}$ performed the best among the four CSD models developed in this study. However, satellite retrieval of $a_{ph}(\lambda)$ in optically complex waters amplifies uncertainty in retrieving $\eta_{MDLsat}$ for the CSD models exploiting $a_{ph}(\lambda)$. Validation results of the CSD model using the $a_{ph\_QAAobs}(\lambda)$, estimated from $R_{rs\_obs}(\lambda)$ through the modified QAA, showed diminished performance, especially for the CSD model$_{SVM-\hat{a}_{ph}(\lambda)}$ (Figure 5f). In this sense, the best-performing model for applications with $R_{rs\_sat}(\lambda)$ is CSD model$_{LR-\hat{R}_{rs}(\lambda)}$. The CSD model$_{LR-\hat{R}_{rs}(\lambda)}$ yielded statistical measures of 0.21 and 1.16 for MAE and bias, respectively. Out of the 53 samples in the validation dataset, estimates for 44 samples (i.e., 83%) were within ±35% of the *in situ* measured values. The associated average and median percent errors with respect to *in situ* values were 28.0% and 16.2%, respectively.

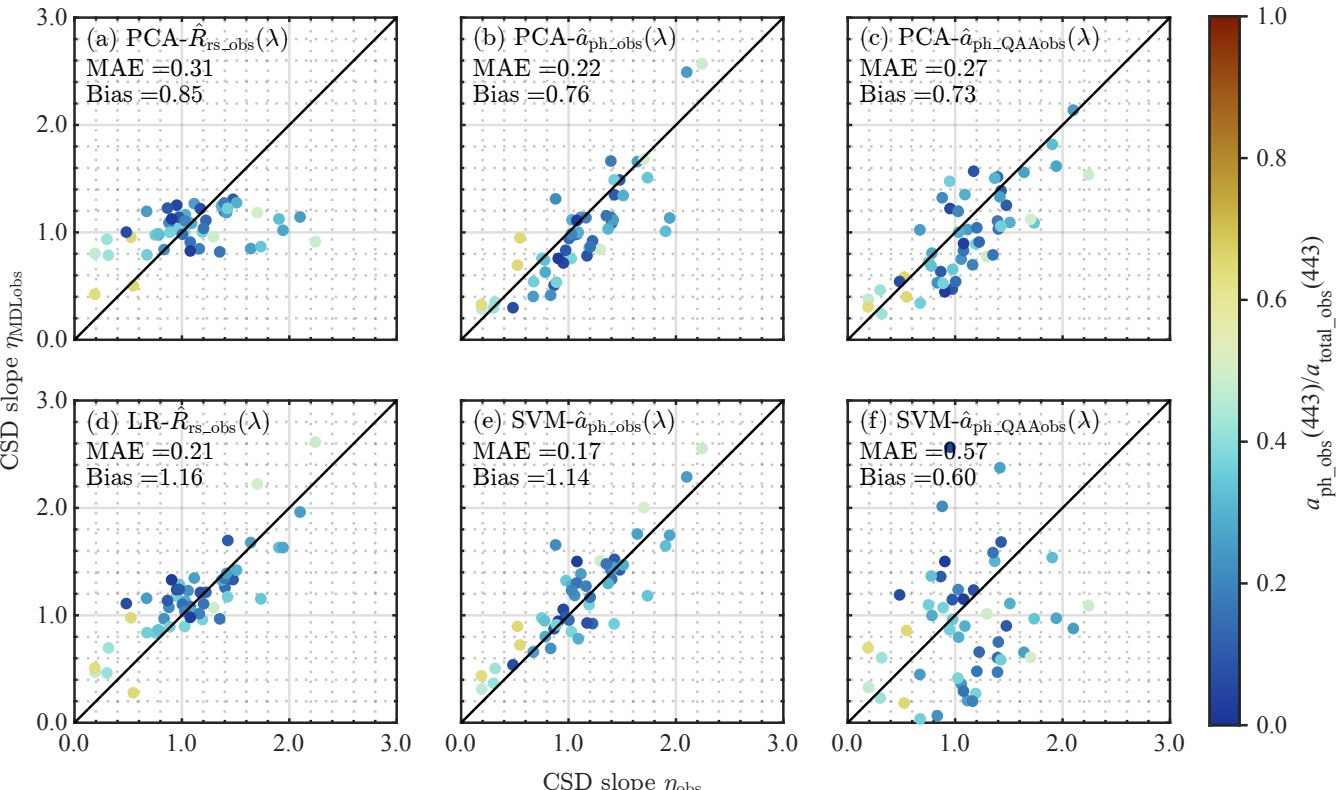

Figure 5. Validation results of the developed models. Comparison between measured ($\eta_{obs}$) and model-estimated $\eta$ values ($\eta_{MDLobs}$) with respect to the fractional contribution of $a_{ph\_obs}(443)$ to $a_{total\_obs}(443)$. Upper (a–c) and lower panels (d–f) show CSD models developed by PCA- and ML-based approaches, respectively. Panels (c) and (f) show the results of the same CSD models in panels (b) and (e) but using $\hat{a}_{ph\_QAAobs}(\lambda)$, whereas panels (b) and (c) use $\hat{a}_{ph\_obs}(\lambda)$ determined from *in situ* observations. MAE denotes the median absolute error.

### 3.7 CSD slope distribution in the Pacific Arctic

Seasonal variations in climatological $\eta_{MDLsat}$ distribution derived by the CSD model$_{LR-\hat{R}_{rs}(\lambda)}$ from $R_{rs\_sat}(\lambda)$ in the Pacific Arctic are shown in Figure 6. The $\eta_{MDLsat}$ values were persistently low in the western side of the Bering Strait, whereas those on the eastern side were generally high throughout the season. Such west-east contrast was also found on the Bering Sea shelf, with low $\eta_{MDLsat}$ values in the west and high $\eta_{MDLsat}$ values in the east. These spatial dynamics in the $\eta_{MDLsat}$ would likely reflect current patterns in the Pacific Arctic. Indeed, SST shows coincident patterns with such spatial variations in $\eta_{MDLsat}$ values (Figure S5), with relatively higher water temperatures tending to contain higher $\eta_{MDLsat}$ as well. The climatological mean $\eta_{MDLsat}$ in the Pacific Arctic decreased from 1.88 to 1.52 from July to September (Figure 7), suggesting an overall shift from smaller to larger phytoplankton communities over the season. More specifically, the fractional contribution of micro-phytoplankton (pico-phytoplankton) to total phytoplankton biomass changed from 0.04 to 0.13 (0.61 to 0.44) between July and September.

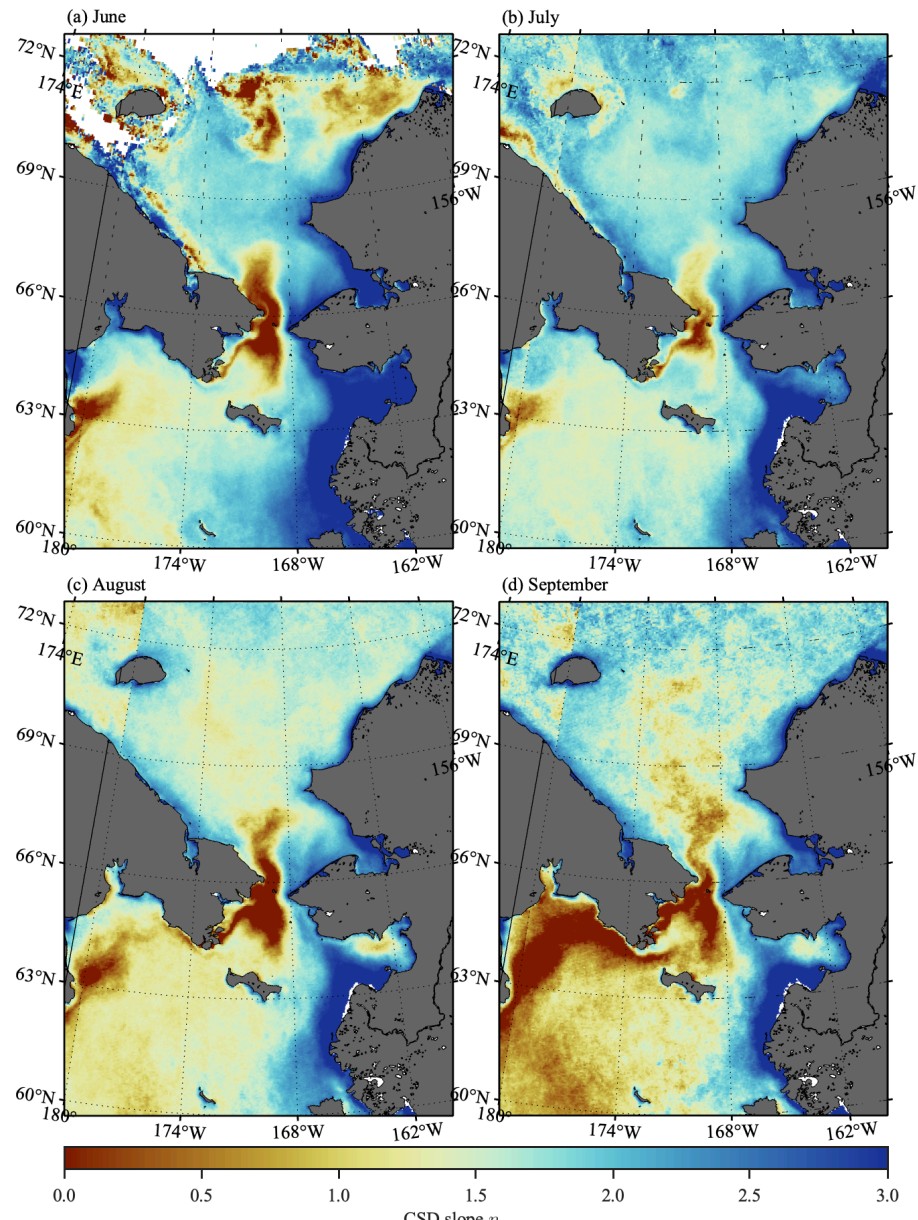

Figure 6. Monthly climatology of $\eta_{\text{MDLsat}}$ values in (a) June, (b) July, (c) August, and (d) September in the Pacific Arctic for
2002–2022 (derived from $R_{\text{rs\_sat}}(\lambda)$ using the CSD model$_{\text{LR}-\hat{R}_{\text{rs}}(\lambda)}$). White areas indicate no valid retrievals due to cloud and/or
sea-ice cover.

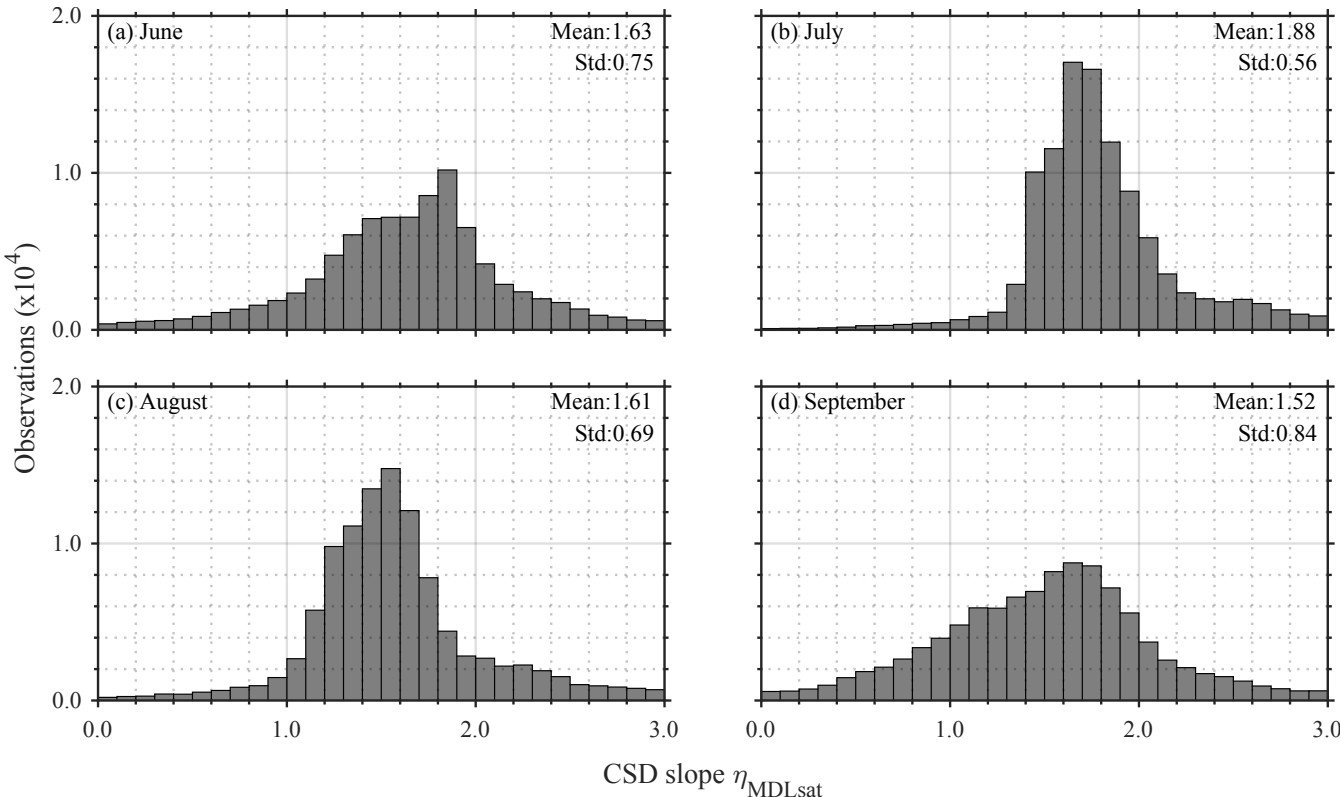

Figure 7. Histograms of monthly climatology of $\eta_{MDLsat}$ values in (a) June, (b) July, (c) August, and (d) September in the Pacific Arctic for 2002–2022.

## 4    Discussion

### 4.1    Taxonomic composition and size structure of phytoplankton community

Numerous studies have reported that the size structure of phytoplankton communities has strong linkages with the taxonomic composition (Finkel et al., 2010). Diatoms and dinoflagellates are generally classified as micro-phytoplankton; prymnesiophytes, chrysophytes, chlorophytes, and cryptophytes are classified as nano-phytoplankton; and prasinophytes and cyanobacteria are grouped into pico-phytoplankton. According to pigment-based taxonomic identification, diatoms and prymnesiophytes were the main phytoplankton taxa contributing to variations in the size structure of the phytoplankton communities (Figure 3b). More specifically, a higher fractional contribution of diatoms was associated with smaller $\eta_{obs}$ values, suggesting a large-sized phytoplankton-dominated condition. In contrast, a higher fractional contribution of prymnesiophytes resulted in larger $\eta_{obs}$ values, indicating a small-sized phytoplankton dominated condition. Overall, shifts in the relative fractions of micro- and nano-size classes drove the change in $\eta_{obs}$, while pico-plankton had less impact.

## 4.2 Responses of optical signatures to phytoplankton size structure

The absolute concentration of phytoplankton pigments in seawater typically affects first-order variability in the magnitude of $R_{rs}(\lambda)$, with secondary impacts on $R_{rs}(\lambda)$ spectral shape associated with diversity in dissolved and particulate properties, such as phytoplankton community composition (Ciotti et al., 2002). Therefore, spectral variations in the magnitude-normalized $\hat{R}_{rs}(\lambda)$ can be reasonably assumed to coincide with changes in the size structure of the phytoplankton community. Indeed, the spectral shape of $\hat{R}_{rs\_obs}(\lambda)$ showed a transition of the peak wavelength from green to blue with increasing $\eta_{obs}$ values (Figure 4b). Likewise, the magnitude of $a_{ph}(\lambda)$ is related to pigment composition and concentration, whereas size information is contained in the shape of the absorption spectrum due to pigment packaging within cells (Bricaud and Morel, 1986b). For example, we found a sharp absorption peak in $\hat{a}_{ph\_obs}(\lambda)$ around 443 nm that appeared to be positively correlated with CSD slope (Figure 4d). Overall, our study demonstrated strong influences of the size structure of phytoplankton communities on $\hat{R}_{rs\_obs}(\lambda)$ and $\hat{a}_{ph\_obs}(\lambda)$, as reported in previous studies (Mouw et al., 2017). Although we found clear linkages in the spectral shape of $\hat{R}_{rs\_obs}(\lambda)$ and $\hat{a}_{ph\_obs}(\lambda)$ with $\eta_{obs}$, it is important to note that $\hat{R}_{rs\_obs}(\lambda)$ is influenced not solely by phytoplankton but also by CDOM and NAP. Since Chl$a$ is generally uncorrelated with CDOM and NAP in coastal waters, the combined impact of absorption and scattering by all water constituents on water-leaving radiance likely accounts for the somewhat poorer performance of the remote sensing-based models compared to the *in situ* pigment absorption based ML models.

## 4.3 Comparison of PCA- and ML-based approaches

The output from PCA consisted of two terms: loading factors and PC scores. Loading factors define the rotations of the axes. PC scores are linearly uncorrelated variables that represent the positions of samples in the new rotated axes, and each is the linear combination of original spectra with corresponding loading factors (Wang et al., 2015). The PCA-based approach adopted here assumes that PC scores are correlated with $\eta$ values, yet this assumption would not have been necessarily valid in this study. Indeed, the PCA-based CSD model showed a degraded performance compared to that of the ML-based model particularly for $\hat{R}_{rs\_obs}(\lambda)$, suggesting that the PCA could have added uncertainties in the retrieval of $\eta$. In fact, simple and direct linear regression resulted in better performance of the CSD model in the case of utilizing $\hat{R}_{rs\_obs}(\lambda)$. In addition, the first two PC modes explained about 95% of spectral variations in $\hat{R}_{rs\_obs}(\lambda)$ and $\hat{a}_{ph\_obs}(\lambda)$. This fact suggests that the other two PC modes (i.e., PC modes 3 and 4) contribute little to explaining the entire spectral variation but may have added uncertainties, especially considering the relatively small dataset used in the current study. Note that the PCA-based approach is a dimensionality reduction method often used to reduce the dimensionality of large data sets by transforming a large set of variables into a smaller one that still contains most of the information in the large set (Corte-Real, 2020). In the case of hyperspectral data, the input variables can easily be hundreds of wavelengths, which imposes a significant computational cost. The PCA can aggregate important spectral features into PC scores and may prove beneficial for developing robust remote sensing algorithms based on hyperspectral data. However, because we are using multispectral data with limited number of predictor variables, this potential benefit of PCA is not realized in our study.

While the conventional least square regression has been used for decades in the development of satellite ocean color algorithms (Fujiwara et al., 2011; Hirata et al., 2011; O'Reilly et al., 1998; Waga et al., 2017, 2019a), more complex ML methods are increasingly being applied and many studies have reported their capability for improved ocean color product retrievals (Chen et al., 2019; Hu et al., 2021, 2018). The least square regression is a statistical method that fits a pre-defined equation to specific data. Due to is relative simplicity, it cannot fully extract hidden patterns in data and/or elicit a deep characterization of intricate relationships between a number of interdependent variables (Martens and Dardenne, 1998). However, the ML approach of learning relationships between the input values and the corresponding output values without predefined or explicated equations requires an extensive dataset that covers complex behaviors in the data and a wide range of environmental conditions (Marzban, 2009). Once trained, ML approaches are powerful tools for the fast and efficient processing of large datasets, such as geospatial satellite data (Paul and Huntemann, 2021; Waga et al., 2022).

One of the key findings of this study is that more complex ML approaches (e.g., support vector machine, ensemble, and neural network) do not always produce more effective models than simple ML approaches (e.g., standard linear regression) (Table 4). While more complex models generally perform better than simpler ones (Makridakis et al., 2022), a complicated or flexible model will pose challenges for interpretation and can end up overfitting random effects (i.e., noises) that are unique to the dataset used for training. If these random effects are not present in new data to which the model is applied, then the model can produce incorrect results when it uses relationships developed based on random phenomena in the training dataset. Thus, the limited size of our dataset (i.e., only 177 samples) likely contributed to the poor performance of the complex ML models. Nonetheless, the CSD model trained with a support vector machine was selected as the best model for $\hat{a}_{\text{ph\_obs}}(\lambda)$. This indicates that the poor performance of complex ML approaches for $\hat{R}_{\text{rs\_obs}}(\lambda)$ may also be associated with other regression-related factors (e.g., number of features, classifier hyper-parameter optimization, and number of cross-validation folds) rather than simply the number of samples used for training (Vabalas et al., 2019). One potential explanation for the better performance with the simple linear regression approach for $\hat{R}_{\text{rs\_obs}}(\lambda)$ is that variance in $\hat{R}_{\text{rs\_obs}}(\lambda)$ for each $\eta_{\text{obs}}$ range was larger compared to that of $\hat{a}_{\text{ph\_obs}}(\lambda)$ (Figure 4c). Complex ML approaches applied to $\hat{R}_{\text{rs\_obs}}(\lambda)$ likely introduced errors related to the variance in the relationship between the spectral features and $\eta_{\text{obs}}$, whereas a simple ML approach captured only predominant features with lesser effects of the variance. Finally, we wish to also express that the type of batch approach employed by MATLAB's Machine Learning App is useful for identifying what type of model might perform well for the problem at hand, however it should not be taken as canon as more complex ML approaches often require careful customization and model design.

## 4.4 Methodological uncertainties and limitations

A major challenge of the ML approach, with some exceptions, such as linear regression, is that it is difficult or impossible to derive a mechanistic understanding of the model-predicted relationship between the input and output values (Ray, 2019). For this reason, the ML approaches are sometimes called "black boxes." This lack of transparency can be problematic in

interpreting the results generated by the model (Vollmer et al., 2020; Wachter et al., 2017). While ML approaches have been

employed in numerous fields besides satellite remote sensing, they have not adequately addressed the issue of causality, which is essential to support wider dissemination and acceptance of the proposed models (Hall et al., 2022). What can be said at this point is that the selection of an ML approach carries with it trade-offs between accuracy and interpretability. Establishing procedures for interpreting how ML models learn and arrive at answers is crucial to not only selecting the appropriate model approach but also for improving reliability and building confidence in the selected approach.

The superior in-situ performance of $a_{ph}(\lambda)$-based models reflects a stronger physical coupling between $\eta$ and $a_{ph}(\lambda)$ (Figure 5). Our preference for the $R_{rs}(\lambda)$-based model is operational, as it avoids uncertainties due to the inversion of $a_{ph}(\lambda)$ from $R_{rs}(\lambda)$ in optically complex waters and yields reliable retrievals for satellite applications; it should not be taken as evidence that $\eta$ is more fundamentally linked to $R_{rs}(\lambda)$ than to $a_{ph}(\lambda)$. A further explanation for why the $a_{ph}(\lambda)$-based model performed better than the $R_{rs}(\lambda)$-based model pertains to measurement uncertainty related to the temporal and spatial scales of the input observations.

Field data for the $a_{ph}(\lambda)$-based model, including pigments and absorption, were derived from analyses of well-mixed water drawn from relatively small sample volumes of few liters, resulting in high confidence that type and concentration of material analyzed for absorption was similar to the material extracted for pigments. By comparison, *in situ* measurements of radiometry for the computation of $R_{rs}(\lambda)$ were measured away from the ship to avoid effects on the light field, at times that were often offset from water sampling by tens of minutes, and represented signals integrated across thousands of liters of near-surface

ocean water. Therefore, uncertainty regarding sample similarity was far greater for $R_{rs\_obs}(\lambda)$ than for $a_{ph\_obs}(\lambda)$.

Our outcome metric was $\eta$, computed from within-sample size fractions rather than absolute Chl*a*. Prior work (Waga et al., 2017) showed that $\eta$ is insensitive to reasonable choices of pore-size boundaries: percent differences in the resulting $\eta_{obs}$ were under 5% across three different typical Chl*a*$_{size}$ cutoffs (i.e., >20, 2–20, and <2 um; >10, 2–10, and <2 μm; and >20, 5–20/<5 μm). Nevertheless, we acknowledge a small residual uncertainty for cruises that used different filters, which could add noise

in heterogeneous conditions. To assess any such effect, we conducted a sensitivity check that removes cruises with differing pore-size splits (i.e., 2007, 2009, 2010) and compared model ranking and error metrics on the reduced subset. These results are summarized in Table S7, which suggests consistent findings across the entire dataset (Table 4).

Absolute Chl*a* can differ across analytical methods (Wang et al., 2025), yet our modeling targets a dimensionless outcome (i.e., $\eta$) computed from within-sample size fractions rather than absolute concentrations. This proportion-based normalization

places fluorometer and HPLC observations on a common scale and helps mitigate method-specific bias in total Chl*a*. The HPLC-based Chl*a*$_{size}$ subset in our compilation is small, which limits our ability to estimate a stable cross-method offset in $\eta$ or to perform a rigorous calibration. Looking ahead, a targeted cross-calibration, paired fluorometer- and HPLC-based Chl*a*$_{size}$ measurements collected contemporaneously across key water masses, would better quantify any residual method dependence in the retrieval of Chl*a*$_{size}$ and further strengthen future assessments.

Overall, our dataset is heterogeneous in time, space, and methods, which introduces non-exchangeability among samples and elevates the risk of biased validation. We used a standard repeated five-fold cross-validation and an external 30% subset to validate the performance of the developed models, but these procedures do not fully control for grouping by cruise, pore-size scheme, analytical approach, or region. As a result, cross-validated skill may be optimistic if folds inadvertently mix samples that are more similar to each other than to the broader population, and the external split may still reflect historical or regional structure (Stock, 2022; Stock and Subramaniam, 2022). Our purpose here is model ranking rather than precise absolute skill; nevertheless, the uncertainty associated with non-stratified resampling should be borne in mind when interpreting differences among approaches. A more conservative assessment is to partition the data into discrete "blocks" according to certain criteria, which enables the creation of independent training and validation folds using stratified blocking (e.g., temporal and spatial blocks) (Zhang et al., 2023). Such cross-validation strategies are preferable for heterogeneous datasets and are recommended for future work and community benchmarks.

### 4.5 Performance of CSD model in optically complex Pacific Arctic waters

Considering the estimation error associated with the semi-analytical IOP inversion algorithm (i.e., the modified QAA), the CSD model$_{\text{SVM}-\hat{a}_{\text{ph}}(\lambda)}$ contains large uncertainties in the retrieval of $\eta$ (Figure 5). This is primarily because the poor performance of the modified QAA in optically complex waters hampered the $a_{\text{ph}}(\lambda)$ retrieval (Table S2), and estimation errors were propagated to the $\hat{a}_{\text{ph}}(\lambda)$-based CSD model for application to satellite data. In other words, the performance of the $\hat{a}_{\text{ph}}(\lambda)$-based CSD model could be improved if a more accurate IOP inversion algorithm were to be established for optically complex waters. Moreover, hyperspectral satellite sensors, such as the NASA Plankton, Aerosol, Cloud, ocean Ecosystem (PACE) mission's primary sensor, the Ocean Color Instrument (OCI), and the planned Surface Biology and Geology (SBG) and Geostationary Littoral Imaging Radiometer (GLIMR) sensors will have the capability to capture more detailed spectral features of $a_{\text{ph}}(\lambda)$ (Dierssen et al., 2023; Werdell et al., 2018), which will greatly benefit satellite-based monitoring of phytoplankton communities (Isada et al., 2015).

Considering that the accuracy goal for satellite-derived Chl*a* is defined as within ±35% of the true value (Hooker and McClain, 2000), and a variety of ocean color products, such as primary productivity (Behrenfeld and Falkowski, 1997), utilize Chl*a* as one of the input parameters, we conclude that the CSD model developed in this study performs sufficiently well in the Pacific Arctic, presuming adequate correction for atmospheric effects in the satellite data. Since this region receives a large amount of freshwater containing CDOM and NAP delivered from rivers (Matsuoka et al., 2007), it was expected that the performance of the CSD model relying on $\hat{R}_{\text{rs}}(\lambda)$ would be influenced by CDOM and NAP, which often dominate the optical properties of seawaters in this region (Chaves et al., 2015; Mustapha et al., 2012; Wang and Cota, 2003). However, the validation results suggest that the CSD model$_{\text{LR}-\hat{R}_{\text{rs}}(\lambda)}$ performed with consistent accuracy regardless of the fractional contribution of $a_{\text{ph\_pbs}}(\lambda)$ to $a_{\text{total\_obs}}(\lambda)$ at 443 nm (Figure 5).

## 4.6    Distribution of CSD slope in the Pacific Arctic

The Pacific Arctic, with a large continental shelf extending from the northern Bering Sea to the southern Chukchi Sea and northwards, has been characterized by a tight pelagic-benthic coupling (Grebmeier et al., 1988, 1989; Grebmeier and McRoy, 1989), with up to 70% of primary production ultimately reaching the seafloor (Walsh et al., 1989). The seasonal cycle of sea-ice formation and melting provides suitable conditions for phytoplankton growth (Stabeno et al., 2010), with large spring diatom blooms occurring at the marginal ice edge and under the ice (Laney and Sosik, 2014; Waga et al., 2021b). The northern Bering and Chukchi Seas are reported to have the highest sinking particulate organic carbon fluxes (0.8–2.5 g C m$^{-2}$ d$^{-1}$) within the world ocean, and the particles collected by moored sediment traps consist of aggregates composed of diatoms exclusively (O'Daly et al., 2020). On the continental shelves in the Pacific Arctic, much of the organic carbon produced in the euphotic layer is directly transported to the seafloor with little or no grazing by zooplankton (Campbell et al., 2009). This strong pelagic-benthic coupling has maintained areas of persistently high benthic biomass, also called benthic hotspots (Grebmeier et al., 2015a), which serve as important foraging areas for upper trophic level benthivores, such as bearded seals, walrus, gray whales, and diving seabirds (Grebmeier, 2006). These hotspots are supported by influxes of organic carbon introduced by vertical transport from the overlying water column and lateral advection (Grebmeier et al., 2015b). Regarding the vertical transport of organic carbon, Waga et al. (2019a) reported that the size structure of phytoplankton communities has a significant relationship with Chl$a$ concentration in the underlying seafloor sediments, suggesting a connection between phytoplankton cell size and benthic macrofaunal biomass in this region.

We found clear spatial variation in the distribution of $\eta_{MDLsat}$ in the Pacific Arctic (Figure 6). For example, on the Bering Sea shelf, the Siberian coast exhibited smaller $\eta_{MDLsat}$ values, whereas larger values were found along the Alaskan coast. Throughout the seasons, there were west-east gradients showing smaller and larger $\eta_{MDLsat}$ values on the Siberian and Alaskan sides of the Bering Strait, respectively. Since a small CSD slope represents a greater proportion of larger-sized phytoplankton, this result indicates larger-sized phytoplankton typically dominated along the Siberian coast, and smaller-sized phytoplankton dominated along the Alaskan coast. In the Pacific Arctic, three major water masses prevail: i.e., the Alaskan Coastal Water, Anadyr Water, and Bering Shelf Water (Coachman et al., 1976; Danielson et al., 2017). The Alaskan Coastal Water is identified with relatively high temperatures and low salinity due to freshwater input flows along the western coast of Alaska out to the Beaufort Sea (Coachman et al., 1976). The Anadyr Water, which flows along the eastern coast of Siberia, has low temperatures and high salinity, and supplies large amounts of nutrients to the Bering Sea and Bering Strait (Coachman et al., 1976). The Bering Shelf Water flows between Anadyr Water and Alaskan Coastal Water on the Bering Sea shelf and forms as these two water masses mix as they pass through the Bering Strait (Grebmeier et al., 1988). In addition to these general current patterns, satellite images of SST (Figure S5) show distinct signatures of cold-water outcroppings in the western side of the Bering Strait, particularly in July and August. Such signatures were associated with friction between the current and the sea floor (Kawaguchi et al., 2020) and accompanied by upward nutrient flux to the surface from the nutrient-rich bottom layer of Anadyr Water (Nishioka et al., 2021), resulting in smaller $\eta_{MDLsat}$ around the Bering Strait. These water mass distributions matched the spatial

pattern in the $\eta_{\text{MDLsat}}$ in the Pacific Arctic, suggesting a tight relationship between nutrient availability and phytoplankton cell size (Ko et al., 2020; Suzuki et al., 2021).

The $\eta_{\text{MDLsat}}$ values in the Pacific Arctic showed clear seasonal changes from June to September (Figure 7). According to previous studies in this region (Waga et al., 2021b; Waga and Hirawake, 2020), ice-associated spring blooms mature primarily within 20 days after sea-ice retreat and then decay gradually until fall blooms occur. Although the timing and presence/absence of spring and fall blooms largely depend on sea-ice conditions and other factors such as wind forcing (Fujiwara et al., 2018; Nishino et al., 2015), June and July are generally characterized as the post-bloom period and August and September are the typical fall bloom period. Such onset and decay of phytoplankton blooms are strongly linked to the size composition of phytoplankton communities in the Pacific Arctic (Waga and Hirawake, 2020), as shown in seasonal variations in $\eta_{\text{MDLsat}}$ values.

## 5    Conclusions

This study developed a CSD model in optically complex Pacific Arctic waters by employing machine learning methods, which exploit hidden, complex relationships between optical signatures and phytoplankton size composition. Considering the large uncertainties in the inversion of $a_{\text{ph}}(\lambda)$ from satellite-derived $R_{\text{rs}}(\lambda)$, we used $R_{\text{rs}}(\lambda)$ directly as a model input instead of $a_{\text{ph}}(\lambda)$, though $a_{\text{ph}}(\lambda)$ is more directly related to the size composition of phytoplankton communities. Neglecting the estimation errors produced from IOP inversion and considering only remotely sensed radiances and phytoplankton absorption spectra from water samples, the best-performing model among the four CSD models examined in this study was the ML-based model with normalized $a_{\text{ph}}(\lambda)$ spectra used as input ($\text{CSD model}_{\text{SVM}-\hat{a}_{\text{ph}}(\lambda)}$), followed by the ML-based model with $R_{\text{rs}}(\lambda)$ ($\text{CSD model}_{\text{LR}-\hat{R}_{\text{rs}}(\lambda)}$), the PCA-based model with $a_{\text{ph}}(\lambda)$ ($\text{CSD model}_{\text{PCA}-\hat{a}_{\text{ph}}(\lambda)}$), and finally the PCA-based model with $R_{\text{rs}}(\lambda)$ ($\text{CSD model}_{\text{PCA}-\hat{R}_{\text{rs}}(\lambda)}$). Within our dataset, the PCA-based CSD model showed a degraded performance compared to that of the ML-based model for both $\hat{R}_{\text{rs\_obs}}(\lambda)$ and $\hat{a}_{\text{ph\_obs}}(\lambda)$. Although the PCA-based approach assumes that PC scores are correlated with $\eta$ values, this assumption would not have been necessarily valid, particularly for $\hat{R}_{\text{rs\_obs}}(\lambda)$. In addition, this study utilized the first four PC modes as representative for spectral features of $\hat{R}_{\text{rs\_obs}}(\lambda)$ and $\hat{a}_{\text{ph\_obs}}(\lambda)$. The first two PC modes explained about 95% of spectral variations in $\hat{R}_{\text{rs\_obs}}(\lambda)$ and $\hat{a}_{\text{ph\_obs}}(\lambda)$, whereas the latter two modes contributed little to explaining the entire spectral variation but may have added uncertainties associated with the PCA step. Another key finding is that more complex ML approaches do not always produce more effective models than standard linear regression. Indeed, simple linear regression outperformed other ML approaches for $\hat{R}_{\text{rs\_obs}}(\lambda)$, whereas the CSD model developed with support vector machine was selected as the best for $a_{\text{ph}}(\lambda)$. Overall, we found benefits in using ML tools to modify and improve the retrieval accuracy of the previously developed CSD model in the Pacific Arctic. Future innovations in machine learning, satellite (and airborne) ocean color sensor capabilities, and IOP algorithms can further contribute to robust, synoptic remote sensing monitoring of phytoplankton size structure in optically complex waters, such as the Arctic Ocean, where rapid change

is altering the dynamics of phytoplankton with cascading effects on higher trophic levels, ecosystem functioning, and marine
resources.

*Code availability*. The codes for CSD models developed in this study are available on GitHub repository (https://github.com/MatlabCode4CSDmodel).

*Author contribution*. Conceptualization of the study was done by HW; the methodology was established by HW, AF, and TH; validation was done by HW; formal analysis of the data was done by HW; *in situ* data were collected by HW, AF, SGA, DK,
KT, AM, and TH; the original draft was prepared by HW; review and editing was done by all authors; visualization of the results was done by HW; project administration was done by WJM, TH, KS, SIS; funding acquisition was done by HW, WJM, DK, TH, KS, and SIS.

*Competing interests*. Koji Suzuki is a member of the editorial board of Biogeosciences. The authors declare that they have no other conflict of interest.

*Acknowledgements*. We sincerely acknowledge the captains and crews of the T/S *Oshoro-maru*, R/V *Mirai*, and R/V *Ukpik* for their expert guidance and cooperation during the cruises. We also express our gratitude to the staff of JAMSTEC, Marine Work Japan, Ltd., and NASA Goddard Space Flight Center (GSFC), for their support in obtaining and analyzing the data. We appreciate the NASA Distributed Active Archive Center (DAAC) for producing and distributing ocean color data.

*Financial support*. This work was supported by the Ministry of Education, Culture, Sports, Science, and Technology of Japan
(MEXT) through the Green Network of Excellence (GRENE) and the Arctic Challenges for Sustainability (ArCS). This research was also supported by NASA Ocean Biology and Biogeochemistry programs 80NSSC22K1055 and 80NSSC25K7431, European Union's Horizon 2020 research and innovation program (Marie Skłodowska-Curie grant agreement no. 101034309), Grant-in-Aids for JSPS Early-Career Scientists 21K14894, and JST CREST JPMJCR23J4.

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
