# Peer review of "Machine learning for estimating phytoplankton size structure from satellite ocean color imagery in optically complex Pacific Arctic waters"

_EGUsphere, 2025_

## Author Comment (AC1)

**Reviewer #1**

The authors develop and compare chlorophyll-a size-distribution (CSD) models to retrieve  $\eta$  (as an indicator of phytoplankton size structure) for an optically complex sector of the Pacific Arctic. They use an in-situ dataset (>150 stations, 2007–2021) to show that machine learning models outperform the commonly used PCA approach, although a simple linear regression on normalized Rrs appears to perform best for satellite applications.

The study is sound and offers novel and relevant insights. The main conclusions are also supported by the analyses. However, several aspects require clarification and strengthening before publication.

> We are grateful for your careful and constructive assessment, which led us to clarify the methods and streamline the presentation. Our responses to your comments are provided in point-by-point manner below.

**Major comments:**

1. The dataset (N=177) is rather heterogeneous, encompassing different decades, methodologies, and water masses. With fewer than 200 samples and a random 70/30 split, there is a clear risk of bias during validation.

Before 2012, the cruises used different filter pore-size schemes. While the 5  $\mu$ m vs. 2  $\mu$ m cutoff for nanophytoplankton may not introduce major differences, the 20  $\mu$ m vs. 10  $\mu$ m cutoff applied in 2009 and 2010 could significantly affect the microphytoplankton fraction. These three cruises alone account for ~1/3 of the dataset.

I would also be cautious about merging fluorometer-derived Chl-a with HPLC-derived values in such a complex region. Is this necessary, particularly when the latter include only 10 samples? Typically, unless the two methods have been explicitly compared and shown to agree for this dataset, it may be better to exclude the HPLC samples.

Furthermore, the in-situ stations span from ~50°N to 78°N, covering the Bering, Chukchi, and Beaufort seas. This spatial heterogeneity thus likely introduces substantial variability. It is also unclear which cruises and years correspond to which regions, but it is likely that different regions were sampled in different years.

Therefore, I recommend the following:

 Consider using stratified random sampling, based on in-situ chlorophyll-a or another variable indicative of distinct water masses or communities. At the very least, test different splits and assess whether significant differences emerge.

- Try performing cross-validation stratified by cruise or, at least, by pore-size scheme to reduce potential bias.
- Compare model performance after removing cruises with differing pore-size splits (2007, 2009, 2010) and the HPLC data. Although the resulting dataset (N=107) would be smaller, it would likely be more homogeneous and potentially yield more robust models.
- Add a supplementary map showing the distribution of stations color-coded by cruise/year.

> We agree that grouped/stratified cross-validation (e.g., cruise-wise folds) is preferable for heterogeneous datasets. Our analyses were conducted in MATLAB's Regression Learner App which, while well-suited for consistent, side-by-side multi-model comparison, does not provide built-in support for cruise- or method-stratified CV. Because our goal is the relative comparison of PCA- and ML-based CSD models, we emphasize model ranking rather than absolute skill.

Our outcome metric is the CSD slope, which is computed from within-sample relative size fractions rather than absolute Chl-a. As shown in Waga et al. (2017), the CSD slope is insensitive to reasonable choices of pore-size boundaries: the percent difference in CSD slope was <5% between filters using >20, 2–20, <2  $\mu m$  and >10, 2–10, <2  $\mu m$ , and similarly between >20, 2–20, <2  $\mu m$  and >20, 5–20, <5  $\mu m$ . That said, we acknowledge a small residual uncertainty for cruises that used different filters, which could add noise in heterogeneous conditions. To bound any such effect, we conducted a sensitivity check that removes cruises with differing pore-size splits (e.g., 2007, 2009, 2010) and compared model ranking and error metrics on the reduced, more homogeneous subset. These results are summarized in Table S7, which suggests consistent findings with the entire dataset (Table 4).

We agree that absolute Chl-a estimates can vary by method. In our analysis, however, we did not merge absolute Chl-a values across methods. Instead, the CSD framework converted each sample to a dimensionless CSD slope based on the relative (withinsample) size-fractioned Chl-a. This normalization mitigates method-specific biases because it relies on proportions rather than absolute concentrations. Consequently, both fluorometer and HPLC samples were converted to the same standardized metric (CSD slope), and direct cross-method comparison of absolute Chl-a is not required. Since HPLC-based size-fractionated Chla data is quite limited, we could not conduct further analysis to assess the consistency/difference in CSD slope between two methods.

To declare these points, we added a new section 4.4 Methodological uncertainties and limitations.

A companion map color-coded by cruise year is provided in the Supplement Figure S1.

2. For a paper focused on estimating phytoplankton size structure from satellite data, I would have expected a comparison with established PSC/PFT algorithms applied to the insitu dataset, even if brief.

Given that you already have both Rrs and pigment data available, this would be straightforward to implement. For example, models developed by T. Hirata and B. Brewin could be applied and compared against your results. Such a comparison would help to contextualize your findings and highlight the added value of your study.

> We applied the diagnostic pigment analysis (DPA) of Hirata et al. (2011) to our in-situ pigment data and compared the resulting micro/nano/pico fractions with (1) the observed size-fractionated chlorophyll and (2) PSC inferred from our  $\eta$ -based approach (Figure R1). The globally tuned DPA coefficients exhibit significant biases for this Arctic/sub-Arctic dataset, which we attribute to regional differences in pigment composition and community structure. These results suggest that established global PSC/PFT algorithms likely require regionalization to perform optimally in the Pacific Arctic. Given that the objective of this paper is to develop and assess ML-based methods for satellite  $\eta$  retrieval, an exhaustive comparison across multiple PSC/PFT algorithms (e.g., Hirata, Brewin) would dilute that focus. We therefore defer such an intercomparison to future work.

Figure R1. Comparison of (a) micro, (b) nano, and (c) pico fractions.  $F_{obs}$  stands for fractions determined from the observed size-fractionated Chla, whereas  $F_{MDL}$  indicates those estimated from either our method (black plots) or Hirata et al (red plots).

3. I recommend reducing the number of figures and tables in the main text.

Currently, there are 12 figures and 7 tables in total in the main body of the manuscript. This makes the manuscript, although very interesting, dense for the reader. Some of these could be moved to the supplementary material. For instance, Figure 11. Also, consider moving parts of the Methods to the supplementary material to further streamline the manuscript.

- > We agree and have streamlined the main text. We moved ancillary items to the Supplement. The revised manuscript now contains 7 figures and 4 tables in the main body.
- 4. The paper presents monthly climatologies of  $\eta$  from MODIS, but it is unclear why no matchup analysis was conducted to verify that the model performs reliably with satellite data.

While the climatology figures are interesting, uncertainties are substantial in such a complex region. Ideally, the authors should identify in-situ matchups and compare  $\eta$  estimates estimated from L2 MODIS images against their dataset. If this is not feasible, a useful alternative would be to compare climatologies restricted to the period of one or two cruises to provide at least a partial validation.

> We attempted a strict in situ–MODIS matchup but obtained only three usable pairs. Most candidate scenes were excluded by standard quality controls (NASA's standard quality flags), spatial homogeneity checks (5×5 grid tests), and temporal proximity (±3 h) to the station time. Because of this scarcity—and because in situ Rrs is available at every station and is the direct input to our  $\eta$  models—we chose to validate using in situ Rrs rather than a statistically underpowered satellite matchup.

We also clarify that the CSD slope climatologies are provided for spatial context (large-scale patterns), not for quantitative skill assessment. In such a dynamic system (sea-ice extent/retreat timing and water-mass shifts), climatology–station mismatches can be substantial; hence we avoided using monthly climatologies for point-wise matchups.

**Minor comments:**

- 1. Please review the reference list carefully, at least the first two references appear not to exist.
- > Thank you for flagging this. We audited the bibliography and discovered two erroneous entries at the beginning of the list. These have been removed, all remaining references were verified.
- 2. Why was a Random Forest regression model not tested given its popularity and flexibility? I find it hard to accept that it would perform worse than a simple linear regression.
- > Thank you for raising this. In MATLAB's Regression Learner, the "Bagged Trees" preset implements a Random Forest–style ensemble (bootstrap aggregation of decision trees with feature subsampling). Its performance was intermediate in our repeated cross-validation: it ranked 11th among Rrs( $\lambda$ ) models and 7th among aph( $\lambda$ ) models. Configuration details (number of learners, leaf size, feature subsampling) and full metrics are provided in the Supplement (Tables S4–S5).

- 3. For the ML models, the authors state that they used the default settings of the MATLAB Regression Learner App but do not specify what these are. I understand this is a common issue with analyses conducted in proprietary software, but for reproducibility it is essential to explicitly report all settings and parameters required to replicate the analysis.
- > We agree that "default settings" is ambiguous. Our intent was simply to use the five-fold cross-validation, which is the default setting. We have removed that phrase and now state the evaluation procedure explicitly as "To avoid the possibility of missing certain representative samples and/or overfitting the models, repeated five-fold cross-validation (ten repeats) was carried out by randomly dividing the development subset (70% of the entire dataset) into five equally sized sets (or five-folds)."

---

## Author Comment (AC2)

**Reviewer #2**

Review of "Machine learning for estimating phytoplankton size structure from satellite ocean color imagery in optically complex Pacific Arctic waters"

This paper is interesting and timely. It tackles an important gap in Arctic Ocean remote sensing: how to retrieve phytoplankton size structure these optically complex waters. The authors explore machine learning to deal with the high uncertainty in going from Rrs to aph. The main finding, that a simple Rrs-based multivariable linear regression model performs best for satellite applications, is significant.

Overall, the study is well-motivated, the methodology is sound, and the results are clear. Revisions are needed, however, to improve the focus and the clarity.

> Thank you for the insightful comments that sharpened the paper's focus and transparency. A detailed, point-by-point response follows.

**Major comments:**

- 1. The paper is slightly unfocused on its main contribution. Is the novelty in the methodology (new CSD model) or in the application (Arctic  $\eta$  distribution)? The Abstract and Introduction should be revised to make it clear.
- > We thank the reviewer and clarify that the primary novelty is methodological, not the application. We have revised the Abstract to state this explicitly. The paper develops and rigorously benchmarks retrievals of CSD slope directly from  $\text{Rrs}(\lambda)$ , and evaluates ML and  $\text{aph}(\lambda)$ -based alternatives to identify viable non-PCA formulations. The maps of the CSD slope are included only as a brief demonstration of feasibility. The final paragraph of the Introduction already frames the study in this method-first way, so we have not made additional changes there.

Abstract: In this study, we have developed a method based on ML to use remote sensing reflectance  $(R_{rs}(\lambda))$  for directly retrieving  $\eta$ , thus avoiding uncertainties due to the inversion of  $a_{ph}(\lambda)$  from  $R_{rs}(\lambda)$ .

Introduction: The current study aims to (1) parameterize CSD models for the Pacific Arctic using spectral features of  $R_{rs}(\lambda)$  and  $a_{ph}(\lambda)$ , (2) assess satellite algorithm performance using an *in situ* dataset, and (3) compare newly developed models with the previously developed PCA-based CSD model.

2. The paper discusses the trade-off between model accuracy and interpretability, in terms of machine learning methods. Since the Rrs-based linear regression model performed best for application, the authors should emphasize its transparency and robustness. I recommend expanding the Methods with more detail on this model.

- > We appreciate the suggestion. To keep the Methods concise while ensuring full transparency, we added a pointer to the Supplement that provides the complete model parameters of the Rrs-based linear regression model in Table S4. Additionally, we will make the MATLAB codes of the developed CSD model available through GitHub, which help readers reproduce our model.
- 2. The strong performance of the ML model with in situ aph reflects a closer fundamental optical link between  $\eta$  and aph than between  $\eta$  and Rrs (Tables 6 and 7). Please clarify that the choice of the Rrs-based model is a practical solution to inversion limitations in optically complex waters, not an indication of a stronger fundamental relationship.
- > We agree. The higher in-situ skill of the aph–based models reflects a closer optical linkage between  $\eta$  and aph than between  $\eta$  and Rrs (see Table 4). Our selection of the Rrs —based linear regression is therefore a practical choice for satellite application in optically complex waters, where inversions to aph are uncertain, and their errors would propagate to  $\eta$ . We have clarified this in the text and explicitly note that choosing the Rrsbased model does not imply a stronger fundamental relationship than the aph linkage; it reflects operational robustness (fewer assumptions, stable performance, and straightforward uncertainty quantification).

To state this clearly, we revised the Abstract and Section 4.4 as follows:

Abstract: Nevertheless, models using in-situ  $a_{\rm ph}(\lambda)$  yielded better accuracy, reflecting a closer optical linkage between  $\eta$  and  $a_{\rm ph}(\lambda)$  than between  $\eta$  and  $R_{\rm rs}(\lambda)$ . Our choice of an  $R_{\rm rs}(\lambda)$ -based model for satellite application is therefore practical, motivated by the limitations and uncertainty of  $a_{\rm ph}(\lambda)$  inversions in optically complex waters.

The second paragraph in Section 4.4: The superior in-situ performance of aph( $\lambda$ )-based models reflects a stronger physical coupling between  $\eta$  and aph( $\lambda$ ) (Tables 6 and 7). Our preference for the Rrs( $\lambda$ )-based model is operational, as it avoids uncertain aph( $\lambda$ ) inversions in optically complex waters and yields stable retrievals at satellite scale; it should not be taken as evidence that  $\eta$  is more fundamentally linked to Rrs( $\lambda$ ) than to aph( $\lambda$ ).

- 3. The statement that the random selection was performed only once "in order to develop and compare different models using the consistent dataset" (Line 230) is a weakness for ML studies. Please comment on the feasibility of cross-validation or at least repeat the split several times and report mean and standard deviation of the metrics.
- > Thank you for flagging this—our earlier wording was misleading. The 30% subset was a single external test set, held out once and used only for final validation (Figure 6). Model development and comparison were conducted on the remaining 70% using repeated 5-fold cross-validation. We have clarified this in the text and now report cross-validation performance as mean ± SD across repeats, alongside the external-test metrics. This

protocol cleanly separates model development/selection (70%) from final assessment (independent 30%).

In the revised version, we repeated the cross-validation for 10 times. Mean  $\pm$  std across 10 repeats are shown in Table 4, as well as Tables S4 and S5 in the Supplement.

- 4. The paper is comprehensive but could be streamlined:
- a. The Discussion section repeats quantitative findings already shown in the Results. Please trim or merge with Results.
- > We agree and have removed duplicative sentences and folded essential interpretive points into the Results where appropriate. The revised Discussion now focuses on synthesis, limitations, and implications rather than re-reporting numerical results.
- b. Some supporting material (e.g., phytoplankton community groups, Figure 4, Table 3) could be moved to the Supplementary.
- > Implemented. We relocated the requested supporting material to the Supplement. The main text now contains 7 figures and 4 tables, improving readability while preserving full detail in the Supplement.
- c. Figure 5 mainly illustrates optical complexity and could also be moved to the Supplementary.
- > Implemented. Figure 5 now appears in the Supplement, with a pointer in the main text.
- d. Since many models are compared, but not all are equally important, the main text should focus on the most effective model, with detailed comparison tables in the Supplementary.
- > Implemented. The main text now emphasizes the most effective model for application, with concise comparative context. Comprehensive performance tables and model-by-model diagnostics are provided in the Supplement, ensuring transparency without overloading the narrative.

**Minor comments:**

- 1. Line 32: Remove "repeat".
- > Done.
- 2. Line 55. Define "AOPs" at first use.

- > It's defined in the third paragraph in the Introduction: "The ocean color variables used in these spectral approaches are grouped into two categories: apparent optical properties (AOPs, e.g., Rrs( $\lambda$ )) and inherent optical properties (IOPs, e.g., aph( $\lambda$ ))."
- 3. Line 61. Please add relevant references on ocean biogeochemical models that use the CSD slope.
- > Direct assimilation of the CSD slope into ocean biogeochemical models is still emerging; to our knowledge it has not yet been widely implemented.
- 4. Line 84. There are many recent publications applying machine learning in ocean colour remote sensing, e.g., : https://doi.org/10.1016/j.rse.2023.113596 and https://doi.org/10.1016/j.rse.2023.113628, and etc.
- > Added the two suggested Remote Sensing of Environment (2023) papers to the Introduction's ML paragraph.
- 5. I recommend adding a table listing all symbols and abbreviations used in the paper for clarity.
- > Implemented. We added a list of definitions and units of symbols used in the manuscript as Table 1.